# Towards Problem-dependent Optimal Learning Rates

**Yunbei Xu**
Columbia University
New York, NY 10027
yunbei.xu@gsb.columbia.edu

**Assaf Zeevi**
Columbia University
New York, NY 10025
assaf@gsb.columbia.edu

## Abstract

We study problem-dependent rates, i.e., generalization errors that scale tightly with the variance or the effective loss at the "best hypothesis." Existing uniform convergence and localization frameworks, the most widely used tools to study this problem, often fail to simultaneously provide parameter localization and optimal dependence on the sample size. As a result, existing problem-dependent rates are often rather weak when the hypothesis class is "rich" and the worst-case bound of the loss is large. In this paper we propose a new framework based on a "uniform localized convergence" principle. We provide the first (moment-penalized) estimator that achieves the optimal variance-dependent rate for general "rich" classes; we also establish improved loss-dependent rate for standard empirical risk minimization.

## 1   Introduction

**Problem Statement.** Consider the following statistical learning setting. Assume that a random sample $z$ follows an unknown distribution $\mathbb{P}$ with support $\mathcal{Z}$. For each realization of $z$, let $\ell(\cdot; z)$ be a real-valued *loss function*, defined over the *hypothesis class* $\mathcal{H}$. Let $h^* \in \mathcal{H}$ be the optimal hypothesis that minimizes the *population risk*

$$\mathbb{P}\ell(h; z) := \mathbb{E}[\ell(h; z)].$$

Given $n$ i.i.d. samples $\{z_i\}_{i=1}^n$ drawn from $\mathbb{P}$, our goal, roughly speaking, is to "learn" a hypothesis $\hat{h} \in \mathcal{H}$ that makes the *generalization error*

$$\mathscr{E}(\hat{h}) := \mathbb{P}\ell(\hat{h}; z) - \mathbb{P}\ell(h^*; z)$$

as small as possible. This pursuit is ubiquitous in machine learning, statistics and stochastic optimization.

Let $\mathcal{V}^*$ and $\mathcal{L}^*$ be the variance and the "effective loss" at the best hypothesis $h^*$:

$$\mathcal{V}^* := \text{Var}[\ell(h^*; z)], \quad \mathcal{L}^* := \mathbb{P}[\ell(h^*; z) - \inf_{\mathcal{H}} \ell(h; z)].$$

We study finite-sample generalization errors that scale tightly with $\mathcal{V}^*$ or $\mathcal{L}^*$, which we call *problem-dependent rates*, without invoking strong convexity or margin conditions. While the direct dependence of $\mathscr{E}(\hat{h})$ on the sample size $n$ is often well-understood, it typically only reflects an "asymptotic" perspective, placing less emphasis on the scale of problem-dependent parameters $\mathcal{V}^*$ and $\mathcal{L}^*$.

**Main challenges.** In absence of strong convexity and margin conditions, perhaps the most popular framework to study problem-dependent rates is the traditional "local Rademacher complexity" analysis [2, 7, 17], which has become a standard tool in learning theory. However, as we will discuss later, this analysis makes the "direct dependence" on the sample size $(n)$ sub-optimal for all "rich" classes with the exception of parametric classes.

The absence of more precise localization analysis also challenges the design of more refined estimation procedures. For example, designing estimators to achieve variance-dependent rates requires penalizing

the empirical second moment to achieve the "right" bias-variance trade-off. Most antecedent work is predicated on either the traditional "local Rademacher complexity" analysis [13, 4] or coarser approaches [9, 15]. Thus, to the best of our knowledge, the question of optimal problem-dependent rates for general rich classes is still open.

**Contributions.** We introduce a new framework to study localization in statistical learning, dubbed "uniform localized convergence," which simultaneously provides optimal "direct dependence" on the sample size, and correct scaling with problem-dependent parameters. This framework resolves some fundamental limitations of existing localization analysis.

We employ the above ideas to design the first estimator that achieves optimal variance-dependent rates for general function classes. The derivation is based on a novel two-stage procedure that optimally penalizes the empirical (centered) second moment. We also establish improved loss-dependent rates for standard empirical risk minimization, which has computational advantages.

When assuming suitable curvature or margin conditions, much progress on problem-dependent rates has been made under particular formulations, such as supervised learning with strong convexity [11, 12, 8]. Our proposed framework leads to substantial progress in these and several other problem settings, a detailed account can be found in the full version of this work [18].

**Organization.** Section 2 introduces our proposed "uniform localized convergence" principle. Section 3 provides preliminaries. Section 4 presents the loss-dependent rate. Section 5 presents the variance-dependent rate. Section 6 illustrates our findings in two examples: non-parametric classes and VC classes.

## 2 The "uniform localized convergence" principle

### 2.1 The current blueprint

Denote the *empirical risk*

$$\mathbb{P}_n\ell(h; z) := \frac{1}{n}\sum_{i=1}^{n}\ell(h; z_i),$$

and consider the following straightforward decomposition of the generalization error

$$\mathscr{E}(\hat{h}) = (\mathbb{P} - \mathbb{P}_n)\ell(\hat{h}; z) + \left(\mathbb{P}_n\ell(\hat{h}; z) - \mathbb{P}_n\ell(h^*; z)\right) + (\mathbb{P}_n - \mathbb{P})\ell(h^*; z). \tag{2.1}$$

The main difficulty in studying $\mathscr{E}(\hat{h})$ comes from bounding the first term $(\mathbb{P} - \mathbb{P}_n)\ell(\hat{h}; z)$, since $\hat{h}$ depends on the $n$ samples. The simplest approach, which does not achieve problem-dependent rates, is to bound the uniform error

$$\sup_{h\in\mathcal{H}}(\mathbb{P} - \mathbb{P}_n)\ell(h; z)$$

over the *entire* hypothesis class $\mathcal{H}$. In order to obtain problem-dependent rates, a natural modification is to consider uniform convergence over *localized* subsets of $\mathcal{H}$.

We first give an overview of the traditional "local Rademacher complexity" analysis [2, 7, 17]. Consider a generic function class $\mathcal{F}$ that we wish to concentrate, which consists of real-valued functions defined on $\mathcal{Z}$ (e.g., one can set $f(z) = \ell(h; z)$). Denote

$$\mathbb{P}f := \mathbb{E}[f(z)], \quad \mathbb{P}_nf := \frac{1}{n}\sum_{i=1}^{n}f(z_i),$$

and denote by $\psi(r; \delta)$ a surrogate function that upper bounds the uniform error within a localized region $\{f \in \mathcal{F} : T(f) \leq r\}$, where we call $T : \mathcal{F} \to \mathbb{R}_+$ the "measurement functional". Formally, let $\psi$ be a function that maps $[0, \infty) \times (0, 1)$ to $[0, \infty)$, which possibly depends on the observed samples $\{z_i\}_{i=1}^{n}$. Assume $\psi$ satisfies for arbitrary fixed $\delta, r$, with probability at least $1 - \delta$,

$$\sup_{f\in\mathcal{F}:T(f)\leq r}(\mathbb{P} - \mathbb{P}_n)f \leq \psi(r; \delta). \tag{2.2}$$

By default we ask $\psi(r; \delta)$ to be a non-decreasing and non-negative function. The main result of the traditional "local Rademacher complexity" analysis can be stated as follows (adapted from [2, Section 3.2]).

**Statement 1** (**current blueprint**). *Assume that $\psi$ is a sub-root function, i.e., $\psi(r; \delta)/\sqrt{r}$ is non-increasing with respect to $r \in \mathbb{R}_+$. Assume the Bernstein condition $T(f) \leq B_e \mathbb{P}[f]$, $B_e > 0$, $\forall f \in \mathcal{F}$. Then with probability at least $1 - \delta$, for all $f \in \mathcal{F}$ and $K > 1$,*

$$(\mathbb{P} - \mathbb{P}_n)f \leq \frac{1}{K}\mathbb{P}f + \frac{100(K-1)r^*}{B_e}, \tag{2.3}$$

*where $r^*$ is the "fixed point" solution of the equation $r = B_e\psi(r; \delta)$.*

Since its inception, Statement 1 has become a standard tool in learning theory. However, it requires a rather technical proof, and it appears to be loose when compared with the original assumption (2.2)—ideally, we would like to directly extend (2.2) to hold uniformly without sacrificing any accuracy. Moreover, some assumptions in the statement are restrictive and might not be necessary.

## 2.2 Key ideas of the "uniform localized convergence" principle.

We provide a surprisingly simple approach which greatly improves and simplifies the current blueprint. While Statement 1 relies heavily on restrictive assumptions like the "sub-root" property of $\psi$ and the Bernstein condition, the following proposition holds essentially without any restrictions.

**Proposition 1** (**uniform localized convergence**). *For function class $\mathcal{F}$ and functional $T : \mathcal{F} \to [0, R]$, assume there is a function $\psi(r; \delta)$, which is non-decreasing with respect to $r$ and satisfies that $\forall \delta \in (0, 1)$, $\forall r \in (0, R]$, with probability at least $1 - \delta$,*

$$\sup_{f \in \mathcal{F}:T(f)\leq r} (\mathbb{P} - \mathbb{P}_n)f \leq \psi(r; \delta). \tag{2.4}$$

*Then, given any $\delta \in (0, 1)$ and $r_0 \in (0, R]$, with probability at least $1 - \delta$, for all $f \in \mathcal{F}$, either $T(f) \leq r_0$ or*

$$(\mathbb{P} - \mathbb{P}_n)f \leq \psi\left(2T(f); \delta(\log_2 \frac{2R}{r_0})^{-1}\right). \tag{2.5}$$

*We note that both $T$ and $\psi$ are allowed to be sample-dependent in the above.*

The key intuition behind Proposition 1 is that the uniform restatement of the "localized" argument (2.4) is nearly cost-free, because the deviations $(\mathbb{P} - \mathbb{P}_n)f$ can be controlled solely by the real valued functional $T(f)$. As a result, we essentially only require uniform convergence over an interval $[r_0, R]$. The "cost" of this uniform convergence, namely, the additional $\log_2(\frac{2R}{r_0})$ term in (2.5), will only appear in the form $\log(\delta/\log_2(\frac{2R}{r_0}))$ in high-probability bounds, which is of a negligible $O(\log \log n)$ order in general.

Formally, we apply a "peeling" technique: we take $r_k = 2^k r_0$, where $k = 1, 2, \ldots, \lceil\log_2 \frac{R}{r_0}\rceil$, and we use the union bound to extend (2.4) to hold for all these $r_k$. Then for any $f \in \mathcal{F}$ such that $T(f) > r_0$ is true, there exists a non-negative integer $k$ such that $2^k r_0 < T(f) \leq 2^{k+1}r_0$. By the non-decreasing property of the $\psi$ function, we then have

$$(\mathbb{P} - \mathbb{P}_n)f \leq \psi\left(r_{k+1}; \delta(\log_2 \frac{2R}{r_0})^{-1}\right) \leq \psi\left(2T(f); \delta(\log_2 \frac{2R}{r_0})^{-1}\right),$$

which is exactly (2.5). Interestingly, the proof of the classical result (Statement 1) relies on a relatively heavy machinery that includes more complicated peeling and re-weighting arguments (see [2, Section 3.1.4]). However, that analysis obscures the key intuition that we elucidate under inequality (2.5).

The results presented in this paper essentially originate from the noticeable gap between Proposition 1 and Statement 1, illustrated by the following (informal) conclusion.

**Statement 2** (**improvements over the current blueprint–informal statement**). *Under the assumptions of Statement 1, Proposition 1 provides a strict improvement over Statement 1. In particular, the slower $\psi$ grows, the larger the gap between the two results, and the bounds become identical only when $\psi$ is proportional to $\sqrt{r}$, i.e., when the function class $\mathcal{F}$ is parametric and not "rich."*

Formalizing as well as providing rigorous justification for this conclusion is relatively straightforward: taking the "optimal choice" of $K$ in Statement 1, we can re-write its conclusion as

$$(\mathbb{P} - \mathbb{P}_n)f \leq 20\sqrt{\frac{r^*\mathbb{P}f}{B_e}} - \frac{r^*}{B_e} \quad \text{[Statement 1]},$$

where the right hand side is of order $\sqrt{r^*\mathbb{P}f/B_e}$ when $\mathbb{P}f > r^*/B_e$, and order $r^*/B_e$ when $\mathbb{P}[f] \leq r^*/B_e$. Our result (2.5) is also of order $r^*/B_e$ when $\mathbb{P}f \leq r^*/B_e$. However, for every $f$ such that $\mathbb{P}f > r^*/B_e$, it is straightforward to verify that under the assumptions in Statetment 1,

$$\psi(2T(f);\delta) \leq \psi(2B_e\mathbb{P}f;\delta) \quad \text{[Bernstein condition: } T(f) \leq B_e\mathbb{P}f]$$

$$\leq \frac{\sqrt{2B_e\mathbb{P}f}}{\sqrt{r^*}}\psi(r^*;\delta) \quad [\psi(r;\delta) \text{ is sub-root]}$$

$$\leq \sqrt{\frac{2r^*\mathbb{P}f}{B_e}} \quad [r^* \text{ is the fixed point of } B\psi(r;\delta)]. \tag{2.6}$$

Therefore, the argument $\psi(2T(f);\delta) \leq \sqrt{2r^*\mathbb{P}f/B_e}$ established by (2.6) shows that the "uniform localized convergence" argument (2.5) strictly improves over Statement 1.

## 3 Preliminaries

Our results on problem-dependent rates essentially only require the loss function to be uniformly bounded by $[-B, B]$, i.e., $|\ell(h;z)| \leq B$ for all $h \in \mathcal{H}$ and $z \in \mathcal{Z}$. This is a standard assumption used in almost all previous works that do not invoke curvature conditions or rely on other problem-specific structure. Extensions to unbounded targets can be obtained via truncation techniques (see, e.g. [5]), and our problem-dependent results allow $B$ to be very large, potentially scaling with $n$.

We represent the complexity through a surrogate function $\psi(r;\delta)$ that satisfies for all $\delta \in (0,1)$,

$$\sup_{f \in \mathcal{F}:\mathbb{P}[f^2] \leq r} (\mathbb{P} - \mathbb{P}_n)f \leq \psi(r;\delta), \tag{3.1}$$

with probability at least $1 - \delta$, where $\mathcal{F}$ is taken to be the *excess loss class*

$$\ell \circ \mathcal{H} - \ell \circ h^* := \{z \mapsto \ell(h;z) - \ell(h^*;z) : h \in \mathcal{H}\}. \tag{3.2}$$

To achieve non-trivial complexity control (and ensure existence of the fixed point), we only consider "meaningful" surrogate functions stated below.

**Definition 1 (meaningful surrogate function).** A bivariate function $\psi(r;\delta)$ defined over $[0,\infty) \times (0,1)$ is called a meaningful surrogate function if it is non-decreasing, non-negative and bounded with respect to $r$ for every fixed $\delta \in (0,1)$.

We note that the above does not place significant restrictions on the choice of the surrogate function: the left hand side of (3.1) is itself non-decreasing and non-negative; and the boundedness requirement can always be met by setting $\psi(r;\delta) = \psi(4B^2;\delta)$ for all $r \geq 4B^2$. We now give the formal definition of fixed points.

**Definition 2 (fixed point).** Given a non-decreasing, non-negative and bounded function $\varphi(r)$ defined over $[0,\infty)$, we define the fixed point of $\varphi(r)$ to be $\sup\{r > 0 : \varphi(r) \leq r\}$. Equivalently, the fixed point of $\varphi(r)$ is the maximal solution to the equation $\varphi(r) = r$.

Given a bounded class $\mathcal{F}$, empirical process theory provides a general way to construct surrogate function by upper bounding the "local Rademacher complexity" $\mathfrak{R}\{f \in \mathcal{F} : \mathbb{P}[f^2] \leq r\}$ (see Lemma 4 in Appendix H). We give the definition of Rademacher complexity for completeness.

**Definition 3 (Rademacher complexity).** For a function class $\mathcal{F}$ that consists of mappings from $\mathcal{Z}$ to $\mathbb{R}$, define

$$\mathfrak{R}\mathcal{F} := \mathbb{E}_{z,\upsilon} \sup_{f \in \mathcal{F}} \frac{1}{n} \sum_{i=1}^{n} \upsilon_i f(z_i), \quad \mathfrak{R}_n\mathcal{F} := \mathbb{E}_{\upsilon} \sup_{f \in \mathcal{F}} \frac{1}{n} \sum_{i=1}^{n} \upsilon_i f(z_i),$$

as the *Rademacher complexiy* and the *empirical Rademacher complexity* of $\mathcal{F}$, respectively, where $\{\upsilon_i\}_{i=1}^{n}$ are i.i.d. Rademacher variables for which $\text{Prob}(\upsilon_i = 1) = \text{Prob}(\upsilon_i = -1) = \frac{1}{2}$. $\mathbb{E}_z$ means taking expectations over $\{z_i\}_{i=1}^{n}$ and $\mathbb{E}_\upsilon$ means taking expectations over $\upsilon_{i=1}^{n}$.

Furthermore, Dudley's integral bound (Lemma 3 in Appendix H) provides one general solution to construct a *computable* upper bound of local Rademacher complexity via the covering number of $\mathcal{F}$. We give the definition of covering number.

**Definition 4 (covering number and metric entropy).** A $\varepsilon-$cover of a function class $\mathcal{F}$ with the $L_2(\mathbb{P}_n)$ metric is a set $\{f_1, \ldots, f_m\} \subseteq \mathcal{F}$ that satisfies for each $f \in \mathcal{F}$, there exists $i \in \{1, \ldots, m\}$ such that $\sqrt{\mathbb{P}_n(f(z) - f_i(z))^2} \leq \varepsilon$. The covering number $\mathcal{N}(\varepsilon, \mathcal{F}, L_2(\mathbb{P}_n))$ is the cardinality of the smallest $\varepsilon-$cover. We call $\log \mathcal{N}(\varepsilon, \mathcal{F}, L_2(\mathbb{P}_n))$ the metric entropy.

# 4  Loss-dependent rates via empirical risk minimization

In this section we are interested in loss-dependent rates, which should scale tightly with $\mathcal{L}^* := \mathbb{P}[\ell(h^*; z) - \inf_{\mathcal{H}} \ell(h; z)]$; the best achievable "effective loss" on $\mathcal{H}$. The following theorem characterizes the loss-dependent rate of empirical risk minimization (ERM) via a surrogate function $\psi$, its fixed point $r^*$, the effective loss $\mathcal{L}^*$ and $B$.

**Theorem 1** (**loss-dependent rate of ERM**). *For the excess loss class $\mathcal{F}$ in (3.2), assume there is a meaningful surrogate function $\psi(r; \delta)$ that satisfies $\forall \delta \in (0, 1)$ and $\forall r > 0$, with probability at least $1 - \delta$,*

$$\sup_{f \in \mathcal{F}: \mathbb{P}[f^2] \leq r} (\mathbb{P} - \mathbb{P}_n) f \leq \psi(r; \delta).$$

*Then the empirical risk minimizer $\hat{h}_{\mathrm{ERM}} \in \arg\min_{\mathcal{H}} \{\mathbb{P}_n \ell(h; z)\}$ satisfies for any fixed $\delta \in (0, 1)$ and $r_0 \in (0, 4B^2)$, with probability at least $1 - \delta$,*

$$\mathscr{E}(\hat{h}_{\mathrm{ERM}}) \leq \psi\left(24B\mathcal{L}^*; \frac{\delta}{C_{r_0}}\right) \vee \frac{r^*}{6B} \vee \frac{r_0}{24B},$$

*where $C_{r_0} = 2\log_2 \frac{8B^2}{r_0}$, and $r^*$ is the fixed point of $6B\psi\left(8r; \frac{\delta}{C_{r_0}}\right)$.*

**Remarks.**  1) The term $r_0$ is negligible since it can be arbitrarily small. One can simply set $r_0 = B^2/n^4$, which will much smaller than $r^*$ in general ($r^*$ is at least of order $B^2 \log \frac{1}{\delta}/n$ in the traditional "local Rademacher complexity" analysis). In high-probability bounds, $C_{r_0}$ will only appear in the form $\log(C_{r_0}/\delta)$, which is of a negligible $O(\log\log n)$ order, so $C_{r_0}$ can be viewed an absolute constant for all practical purposes. As a result, our generalization error bound can be viewed to be of the order

$$\mathscr{E}(\hat{h}_{\mathrm{ERM}}) \leq O\left(\psi(B\mathcal{L}^*; \delta) \vee \frac{r^*}{B}\right). \tag{4.1}$$

2) By using the empirical "effective loss," $\mathbb{P}_n[\ell(\hat{h}_{\mathrm{ERM}}; z) - \inf_{\mathcal{H}} \ell(h; z)]$, to estimate $\mathcal{L}^*$, the loss-dependent rate can be estimated from data without knowledge of $\mathcal{L}^*$. We defer the details to Appendix A.

**Comparison to existing results.** Under additional restrictions (to be explained later), the traditional analysis (2.3) leads to a loss-dependent rate of the order [2]

$$\mathscr{E}(\hat{h}_{\mathrm{ERM}}) \leq O\left(\sqrt{\frac{\mathcal{L}^* r^*}{B}} \vee \frac{r^*}{B}\right), \tag{4.2}$$

which is strictly worse than our result (4.1) due to reasoning following Statement 2. When $B\mathcal{L}^* \leq O(r^*)$, both (4.1) and (4.2) are dominated by the order $r^*/B$ so there is no difference between them. However, when $B\mathcal{L}^* \geq \Omega(r^*)$, our result (4.1) will be of order $\psi(B\mathcal{L}^*; \delta)$ and the previous result (4.2) will be of order $\sqrt{\mathcal{L}^* r^*/B}$. In this case, the square-root function $\sqrt{\mathcal{L}^* r^*/B}$ is only a coarse relaxation of $\psi(B\mathcal{L}^*; \delta)$: as the traditional analysis requires $\psi$ to be sub-root, we can compare the two orders by

$$\psi\left(B\mathcal{L}^*; \delta\right) \overset{\text{sub-root}}{\leq} \sqrt{\frac{B\mathcal{L}^*}{r^*}} \psi(r^*; \delta) \overset{\text{fixed point}}{=} O\left(\sqrt{\frac{\mathcal{L}^* r^*}{B}}\right). \tag{4.3}$$

The "sub-root" inequality (the first inequality in (4.3)) becomes an equality when $\psi(r; \delta) = O(\sqrt{dr/n})$ in the parametric case, where $d$ is the parametric dimension. However, when $\mathcal{F}$ is rich, $\psi(r; \delta)/\sqrt{r}$ will be strictly decreasing so that the "sub-root" inequality can become quite loose. For example, when $\mathcal{F}$ is a non-parametric class we often have $\psi(r; \delta) = O(\sqrt{r^{1-\rho}/n})$ for some $\rho \in (0, 1)$. The richer $\mathcal{F}$ is (e.g., the larger $\rho$ is), the looser the "sub-root" inequality. This intuition will be validated via examples in Section 6.

Theorem 1 also applies to broader settings than previous results. For example, in [2] it is assumed that the loss is non-negative, and their original result only adapts to $\mathbb{P}\ell(h^*; z)$ rather than the "effective loss" $\mathcal{L}^*$. Our proof (see Appendix D) is quite different as we bypass the Bernstein condition (which is traditionally implied by non-negativity, but not satisfied by the class used here), bypass the sub-root assumption on $\psi$, and adapt to the "better" parameter $\mathcal{L}^*$.

# 5 Variance-dependent rates via moment penalization

The loss-dependent rate proved in Theorem 1 contains a complexity parameter $B\mathcal{L}^*$ within its $\psi$ function, which may still be much larger than the optimal variance $\mathcal{V}^*$. Despite its prevalent use in practice, standard empirical risk minimization is unable to achieve variance-dependent rates in general. An example is given in [13] where $\mathcal{V}^* = 0$ and the optimal rate is at most $O(\log n/n)$, while $\mathscr{E}(\hat{h}_{\mathrm{ERM}})$ is proved to be slower than $n^{-\frac{1}{2}}$.

We follow the path of penalizing empirical second moments (or variance) [9, 15, 13, 4] to design an estimator that achieves the "right" bias-variance trade-off for general, potentially "rich," classes. Our proposed estimator simultaneously achieves correct scaling on $\mathcal{V}^*$, along with minimax-optimal sample dependence ($n$). Besides empirical first and second moments, it only depends on the boundedness parameter $B$, a computable surrogate function $\psi$, and the confidence parameter $\delta$. All of these quantities are essentially assumed known in previous works: e.g., [9, 15] require covering number of the loss class, which implies a computable surrogate $\psi$ via Dudley's integral bound; and estimators in [13, 4] rely on the fixed point $r^*$ of a computable surrogate $\psi$.

In order to adapt to $\mathcal{V}^*$, we use a sample-splitting two-stage estimation procedure (this idea is inspired by the prior work [4]). Without loss of generality, we assume access to a data set of size $2n$. We split the data set into the "primary" data set $S$ and the "auxiliary" data set $S'$, both of which are of size $n$. We denote $\mathbb{P}_n$ the empirical distribution of the "primary" data set, and $\mathbb{P}_{S'}$ the empirical distribution of the "auxiliary" data set.

**Strategy 1** (**the two-stage sample-splitting estimation procedure.**). *At the first-stage, we derive a preliminary estimate of $\mathcal{L}_0^* := \mathbb{P}\ell(h^*; z)$ via the "auxiliary" data set $S'$, which we refer to as $\mathcal{L}_{S'}^*$. Then, at the second stage, we perform regularized empirical risk minimization on the "primal" data set $S$, which penalizes the centered second moment $\mathbb{P}_n[(\ell(h; z) - \mathcal{L}_{S'}^*)^2]$.*

As we will present later, it is rather trivial to obtain a qualified preliminary estimate $\mathcal{L}_{S'}^*$ via empirical risk minimization. Therefore, we firstly introduce the second-stage moment-penalized estimator, which is more crucial and interesting.

**Strategy 2** (**the second-stage moment-penalized estimator.**). *Consider the excess loss class $\mathcal{F}$ in (3.2). Let $\psi(r; \delta)$ be a meaningful surrogate function that satisfies $\forall \delta \in (0, 1)$, $\forall r > 0$, with probability at least $1 - \delta$,*

$$4\mathfrak{R}_n\{f \in \mathcal{F} : \mathbb{P}_n[f^2] \leq 2r\} + \sqrt{\frac{2r \log \frac{8}{\delta}}{n}} + \frac{9B \log \frac{8}{\delta}}{n} \leq \psi(r; \delta).$$

*Denote $C_n = 4\log_2 n + 10$. Given a fixed $\delta \in (0, 1)$, let the estimator $\hat{h}_{\mathrm{MP}}$ be*

$$\hat{h}_{\mathrm{MP}} \in \arg\min_{\mathcal{H}} \left\{ \mathbb{P}_n\ell(h; z) + \psi\left(16\mathbb{P}_n[(\ell(h; z) - \mathcal{L}_{S'}^*)^2]; \frac{\delta}{C_n}\right) \right\}. \tag{5.1}$$

Given an arbitrary preliminary estimate $\mathcal{L}_{S'}^* \in [-B, B]$, we can prove that the generalization error of the moment-penalized estimator $\hat{h}_{\mathrm{MP}}$ is at most

$$\mathscr{E}(\hat{h}_{\mathrm{MP}}) \leq 2\psi\left(c_0\left[\mathcal{V}^* \vee (\mathcal{L}_{S'}^* - \mathcal{L}_0^*)^2 \vee r^*\right]; \frac{\delta}{C_n}\right), \tag{5.2}$$

with probability at least $1 - \delta$, where $c_0$ is an absolute constant, and $r^*$ is the fixed point of $16B\psi(r; \frac{\delta}{C_n})$. Moreover, the first-stage estimation error will be negligible if

$$(\mathcal{L}_{S'}^* - \mathcal{L}_0^*)^2 \leq O(r^*). \tag{5.3}$$

It is rather elementary to show that performing the standard empirical risk minimization on $S'$ suffices to satisfy (5.3), provided an additional assumption that $\psi$ is a "sub-root" function. We now give our theorem on the generalization error following this two-stage procedure.

**Theorem 2** (**variance-dependent rate**). *Let $\mathcal{L}_{S'}^* = \inf_{\mathcal{H}} \mathbb{P}_{S'}\ell(h; z)$ be attained via empirical risk minimization on the auxiliary data set $S'$. Assume that the meaningful surrogate function $\psi(r; \delta)$ is*

*"sub-root," i.e.* $\frac{\psi(r;\delta)}{\sqrt{r}}$ *is non-increasing over* $r \in [0, 4B^2]$ *for all fixed* $\delta$. *Then for any* $\delta \in (0, \frac{1}{2})$, *by performing the moment-penalized estimator in Strategy 2, with probability at least* $1 - 2\delta$,

$$\mathscr{E}(\hat{h}_{\mathrm{MP}}) \leq 2\psi\left(c_1 \mathcal{V}^*; \frac{\delta}{C_n}\right) \vee \frac{c_1 r^*}{8B},$$

*where* $r^*$ *is the fixed point of* $B\psi(r; \frac{\delta}{C_n})$ *and* $c_1$ *is an absolute constant.*

**Remarks.** 1) In high-probability bounds, $C_n$ will only appear in the form $\log(C_n/\delta))$, which is of a negligible $O(\log \log n)$ order, so there is no much difference to view $C_n$ as an absolute constant.

2) The "sub-root" assumption in Theorem 2 is only used to to bound the first-stage estimation error (see (5.3)). This assumption is not needed for the result (5.2) on the second-stage moment penalized estimator.

3) Replacing $\mathcal{V}^*$ by an empirical centered second moment, we can prove a fully data-dependent generalization error bound that is computable from data without the knowledge of $\mathcal{V}^*$. We leave the full discussion to Appendix A.

**Comparison to existing results.** The best variance-dependent rate attained by existing estimators is of the order [4]

$$\sqrt{\frac{\mathcal{V}^* r^*}{B^2}} \vee \frac{r^*}{B},$$

which is strictly worse than the rate proved in Theorem 2. The reasoning is similar to Statement 2 and the explanation after Theorem 1: when $\mathcal{V}^* \leq O(r^*)$ the two results are essentially identical, but our estimator can perform much better when $\mathcal{V}^* \geq \Omega(r^*)$. Because $\psi$ is sub-root and $r^*$ is the fixed point, we can compare the orders of the rates

$$\psi(\mathcal{V}^*; \delta) \overset{\text{sub-root}}{\leq} \sqrt{\frac{\mathcal{V}^*}{r^*}} \psi(r^*; \delta) \overset{\text{fixed point}}{=} O\left(\sqrt{\frac{\mathcal{V}^* r^*}{B^2}}\right).$$

Since variance-dependent rates are generally used in applications that require robustness or exhibit large worst-case boundedness parameter, $\mathcal{V}^* \geq r^*$ is the more critical regime where one wants to ensure the estimation performance will not degrade.

**Discussion.** Per our "uniform localized convergence" principle, the most obvious difficulty in proving Theorem 2 is in establishing (5.2): the empirical second moment is sample-dependent, whereas standard tools in empirical process theory (e.g., Talagrand's concentration inequality, see Lemma 4) require the localized subsets to be independent of the samples. The core techniques in our proof essentially overcome this difficulty by concentrating data-dependent localized subsets to data-independent ones. This idea may be of independent interest; we defer details to Appendix E.

The tightness of our variance-dependent rates depend on tightness of the computable surrogate function $\psi$. When covering numbers of the excess loss class are given, a direct choice is Dudley's integral bound (Lemma 3 in Appendix H), which is known to be rate-optimal for many important classes.

Previous approaches usually take a simper regularization term [9, 4] that is proportional to the square root of the empirical second moment (or empirical variance). That type of penalization is "too aggressive" for rich classes from our viewpoint. [13] propose a regularization term that preserves convexity of empirical risk. However, based on an equivalence proved in their paper, they have similar limitations to the approaches that penalizes the square root of the empirical variance.

# 6 Discussion and illustrative examples

## 6.1 Discussion

Recall that our loss-dependent rates and variance-dependent (moment-penalized) rates are of the orders

$$\mathscr{E}(\hat{h}_{\mathrm{ERM}}) \leq O\left(\psi(B\mathcal{L}^*; \delta) \vee \frac{r^*}{B}\right) \quad \text{and} \quad \mathscr{E}(\hat{h}_{\mathrm{MP}}) \leq O\left(\psi(\mathcal{V}^*; \delta) \vee \frac{r^*}{B}\right), \tag{6.1}$$

respectively. In contrast, the best known loss-dependent rates [2] and variance-dependent rates [4] are of the orders

$$\mathscr{E}(\hat{h}_{\text{ERM}}) \leq O\left(\sqrt{\frac{\mathcal{L}^* r^*}{B}} \vee \frac{r^*}{B}\right) \quad \text{and} \quad \mathscr{E}(\hat{h}_{\text{previous}}) \leq O\left(\sqrt{\frac{\mathcal{V}^* r^*}{B^2}} \vee \frac{r^*}{B}\right), \qquad (6.2)$$

respectively (we use $\hat{h}_{\text{previous}}$ to denote the previous best known moment-penalized estimator proposed in [4]). To illustrate the noticeable gaps between our new results and previous known ones, we compare the two different variance-dependent rates in (6.1) and (6.2) on two important families of "rich" classes: non-parametric classes of polynomial growth and VC classes. The implications of this comparison will similarly apply to loss-dependent rates.

Before presenting the advantages of the new problem-dependent rates, we would like to discuss how to compute them. In Theorem 1 and Theorem 2, the class of concentrated functions, $\mathcal{F}$, is the excess loss class $\ell \circ \mathcal{H} - \ell \circ h^*$ in (3.2). As we have mentioned in earlier sections, a general solution for the $\psi$ function is to use Dudley's integral bound (Lemma 3 in Appendix H). Knowledge of the metric entropy of the excess loss class can be used to calculate Dudley's integral bound and construct the surrogate function $\psi$ needed in our theorems. Note that there is no difference between the metric entropy of the excess loss class and metric entropy of the loss class itself: from the definition of covering number and metric entropy, one has

$$\log \mathcal{N}(\varepsilon, \ell \circ \mathcal{H} - \ell \circ h^*, L_2(\mathbb{P}_n)) = \log \mathcal{N}(\varepsilon, \ell \circ \mathcal{H}, L_2(\mathbb{P}_n)).$$

We comment that almost all existing *theoretical* works that discuss general function classes and losses [2, 9, 15, 4] impose metric entropy conditions on the loss class/excess loss class rather than the hypothesis class, and we follows that line as well to allow for a seamless comparison of the results. As a complement, we will discuss how to obtain such metric entropy conditions for practical applications in Appendix B.

### 6.2 Non-parametric classes of polynomial growth

**Example 1** (**non-parametric classes of polynomial growth**). Consider a loss class $\ell \circ \mathcal{H}$ with the metric entropy condition

$$\log \mathcal{N}(\varepsilon, \ell \circ \mathcal{H}, L_2(\mathbb{P}_n)) \leq O\left(\varepsilon^{-2\rho}\right), \qquad (6.3)$$

where $\rho \in (0, 1)$ is a constant. Using Dudley's integral bound to find $\psi$ and solving $r \leq O\left(B\psi(r; \delta)\right)$, it is not hard to verify that

$$\psi(r; \delta) \leq O\left(\sqrt{\frac{r^{1-\rho}}{n}}\right), \quad r^* \leq O\left(\frac{B^{\frac{2}{1+\rho}}}{n^{\frac{1}{1+\rho}}}\right).$$

As a result, our variance-dependent rate (6.1) is of the order

$$\mathscr{E}(\hat{h}_{\text{MP}}) \leq O\left(\mathcal{V}^{* \frac{1-\rho}{2}} n^{-\frac{1}{2}} \vee \frac{r^*}{B}\right), \qquad (6.4)$$

which is $O\left(\mathcal{V}^{* \frac{1-\rho}{2}} n^{-\frac{1}{2}}\right)$ when $\mathcal{V}^* \geq \Omega(r^*)$. In contrast, the previous best-known result (6.2) is of the order

$$\mathscr{E}(\hat{h}_{\text{previous}}) \leq O\left(\sqrt{\mathcal{V}^*} B^{-\frac{\rho}{1+\rho}} n^{-\frac{1}{2+2\rho}} \vee \frac{r^*}{B}\right), \qquad (6.5)$$

which is $O\left(\sqrt{\mathcal{V}^*} B^{-\frac{\rho}{1+\rho}} n^{-\frac{1}{2+2\rho}}\right)$ when $\mathcal{V}^* \geq \Omega\left(r^*\right)$. Therefore, for arbitrary choice of $n, \mathcal{V}^*, B$, the "sub-optimality gap" is

$$\text{ratio between (6.5) and (6.4)} := \frac{\sqrt{\mathcal{V}^*} B^{-\frac{\rho}{1+\rho}} n^{-\frac{1}{2+2\rho}} \vee \frac{r^*}{B}}{\mathcal{V}^{* \frac{1-\rho}{2}} n^{-\frac{1}{2}} \vee \frac{r^*}{B}} = 1 \vee (\mathcal{V}^*(\frac{n}{B^2})^{\frac{1}{1+\rho}})^{\frac{\rho}{2}}, \qquad (6.6)$$

which can be arbitrary large and grows polynomially with $n$.

We consider two stylized regimes as follows (we use the notation $\approx$ when the left hand side and the right hand side are of the same order).

- The more "traditional" regime: $B \approx 1$, $\mathcal{V}^* \approx n^{-a}$ where $a > 0$ is a fixed constant. This regime captures the traditional supervised learning problems where $B$ is not large, but one wants to use the relatively small order of $\mathcal{V}^*$ to achieve "faster" rates.
- The "high-risk" regime: $B \approx n^b$ where $b > 0$ is a fixed constant, and $\mathcal{V}^* \ll B^2$ (i.e., $\mathcal{V}^*$ is much smaller than order $n^{2b}$). This regime captures modern "high-risk" learning problems such as counterfactual risk minimization [15], policy learning [1], and supervised learning with limited number of samples. In those settings, the worst-case boundedness parameter is considered to scale with $n$ so that one wants to avoid (or reduce) the dependence on $B$.

In both the two regimes, generalization errors via naive (non-localized) uniform convergence arguments will be worse than our approach by orders polynomial in $n$, so we only need to compare with previous variance-dependent rates.

**The "traditional" regime.** The "sub-optimality gap" (6.6) is $1 \vee (\mathcal{V}^* n^{\frac{1}{1+\rho}})^{\frac{\rho}{2}}$. It is quite clear that when $\mathcal{V}^* \approx n^{-a}$ where $0 < a < \frac{1}{1+\rho}$, our variance-dependent rate improves over all previous generalization error rates by orders polynomial in $n$.

**The "high-risk" regime.** We restrict our attention to the simple case $B^{\frac{2}{1+\rho}} \leq \mathcal{V}^* \ll 4B^2$ to gain some insight, where our result exhibits an improvement of order $O(n^{\frac{\rho}{2}(\frac{1}{1+\rho})})$ relative to the previous result. Clearly the larger $\rho$, the more improvement we provide. By letting $\rho \to 1$ our improvement can be as large as $O(n^{\frac{1}{4}})$.

## 6.3 VC-type classes

Our next example considers VC-type classes. Although this classical example has been extensively studied in learning theory, our results provide strict improvements over antecedents.

**Example 2 (VC-type classes).** One general definition of VC-type classes (which is not necessarily binary) uses the metric entropy condition. Consider a loss class $\ell \circ \mathcal{H}$ that satisfies

$$\log \mathcal{N}(\varepsilon, \ell \circ \mathcal{H}, L_2(\mathbb{P}_n)) \leq O\left(d \log \frac{1}{\varepsilon}\right),$$

where $d$ is th so-called the Vapnik–Chervonenkis (VC) dimension [16]. Using Dudley's integral bound to find the surrogate $\psi$ and solving $r \leq O(B\psi(r; \delta))$, it can be proven [7] that

$$\psi(r; \delta) \leq O\left(\sqrt{\frac{dr}{n} \log \frac{8B^2}{r}} \vee \frac{Bd}{n} \log \frac{8B^2}{r}\right), \quad r^* \leq O\left(\frac{B^2 d \log n}{n}\right).$$

Recently, [4] proposed a moment-penalized estimator whose generalization error is of the rate

$$\mathscr{E}(\hat{h}_{\text{previous}}) \leq O\left(\sqrt{\frac{d\mathcal{V}^* \log n}{n}} + \frac{Bd \log n}{n}\right),$$

in the worst case without invoking other assumptions. This result has a $O(\log n)$ gap compared with the $\Omega(\sqrt{\frac{d\mathcal{V}^*}{n}})$ lower bound [3], which holds for arbitrary sample size. There is much recent interest focused on when the sub-optimal $\log n$ factor can be removed [1, 4].

By applying Theorem 2, our refined moment-penalized estimator gives a generalization error bound of tighter rate

$$\mathscr{E}(\hat{h}_{\text{MP}}) \leq O\left(\sqrt{\frac{d\mathcal{V}^* \log \frac{8B^2}{\mathcal{V}^*}}{n}} \vee \frac{Bd \log n}{n}\right). \tag{6.7}$$

This closes the $O(\log n)$ gap in the regime $\mathcal{V}^* \geq \Omega(\frac{B^2}{(\log n)^\alpha})$, where $\alpha > 0$ is an arbitrary positive constant. Though this is not the central regime, it is the first positive result that closes the notorious $O(\log n)$ gap without invoking any additional assumptions on the loss/hypothesis class (e.g., the rather complex "capacity function" assumption introduced in [4]). We anticipate additional improvements are possible under further assumptions on the hypothesis class and the loss function.

## Broader Impact

Problem-dependent rates are relevant to many practical aspects of machine learning. Our theory may be useful to the design of estimators and the certification of confidence intervals in causal inference and counterfactual learning problems, as the focus there is typically on avoiding worst-case parameter-dependence (see also discussion on "counterfactual risk minimization" in Appendix B). We derive improved loss-dependent guarantees for empirical risk minimization, one of the most widely used methods for constructing estimators in machine learning practice. It is possible that our strategy penalizing the empirical variance may find applications in fairness, as variance penalized estimators have been shown to be effective in such problems [6]. Having said that, this work is not expected to have any direct societal consequence absent further adaption to specific applications.

## Funding Disclosure

This work did not receive third party funding or third party support.

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
