[Supplementary Material]

## Supplementary material

**Full version.** We refer to the full version of this work [18] (arXiv: 2011.06186) for more general results and more complete proofs.

**Organization.** In Appendix A we will present problem-dependent bounds that can be computed from data, where the unknown quantity $\mathcal{L}^*$ and $\mathcal{V}^*$ are replaced by some empirical estimates. In Appendix B we will discuss several application areas of our theoretical results. In Appendix C-H, we will present proofs of the theoretical results.

## A  Estimating the loss-dependent and variance-dependent rates from data

In the remarks following Theorem 1, we comment that fully data-dependent loss-dependent bounds can be derived using the empirical "effective loss," $\mathbb{P}_n[\ell(\hat{h}_{\mathrm{ERM}}; z) - \inf_{\mathcal{H}} \ell(h; z)]$ to estimate the unknown parameter $\mathcal{L}^*$. Here we present the full details and some discussion of this approach.

**Corollary 3 (estimate of the loss-dependent rate from data).** *Recall the term $\mathcal{L}^*$ is $\mathbb{P}[\ell(h^*; z) - \inf_{\mathcal{H}} \ell(h^*; z)]$ and denote $\widehat{\mathcal{L}^*} = \mathbb{P}_n[\ell(\hat{h}_{\mathrm{ERM}}; z) - \inf_{\mathcal{H}} \ell(h; z)]$. Under the conditions of Theorem 1, setting $C_n = 2 \log_2 n + 6$, then for any fixed $\delta \in (0, \frac{1}{2})$, with probability at least $1 - 2\delta$, we have*

$$\mathscr{E}(\hat{h}_{\mathrm{ERM}}) \leq \psi\left( cB\widehat{\mathcal{L}^*}; \frac{\delta}{C_n} \right) \vee \frac{cr^*}{B} \vee \frac{cB \log \frac{2}{\delta}}{n} \tag{A.1}$$

*and*

$$\mathcal{L}^* \leq c_1 \left( \widehat{\mathcal{L}^*} \vee \frac{r^*}{B} \vee \frac{B \log \frac{2}{\delta}}{n} \right) \leq c_2 \left( \mathcal{L}^* \vee \frac{r^*}{B} \vee \frac{B \log \frac{2}{\delta}}{n} \right), \tag{A.2}$$

*where $c, c_1, c_2$ are absolute constants.*

**Remarks.**  1) The $B \log \frac{2}{\delta}/n$ terms (A.1) and (A.2) are negligible, because $r^*$ is at least of order $B^2 \log \frac{1}{\delta}/n$ for most practical applications. This order is unavoidable in traditional "local Rademacher complexity" analysis and two-sided concentration inequalities.

2) The generalization error bound (A.1) shows that without knowledge of $\mathcal{L}^*$, one can estimate the order of our loss-dependent rate by using $\widehat{\mathcal{L}^*} = \mathbb{P}_n[\ell(\hat{h}_{\mathrm{ERM}}; z) - \inf_{\mathcal{H}} \ell(h; z)]$ as a proxy. Despite replacing $\mathcal{L}^*$ by $\widehat{\mathcal{L}^*}$, other quantities in the bound remain unchanged in order.

3) The inequality (A.2) shows that the estimation of $\mathcal{L}^*$ is tight.

The proof of Corollary 3 can be found in Appendix F.

In the remark following Theorem 2, we comment that fully data-dependent variance-dependent bounds can be derived by employing an empirical estimate to the unknown parameter $\mathcal{V}^*$. Here we present the full details and some discussion of this approach.

**Corollary 4 (estimate of the variance-dependent rate from data).** *Consider the empirical centered second moment*

$$\widehat{\mathcal{V}^*} := \mathbb{P}_n \left[ \ell(\hat{h}_{\mathrm{NMP}}; z) - \widehat{\mathcal{L}_0^*})^2 \right],$$

*where $\mathcal{L}_{S'}^* \in [-B, B]$ is the preliminary estimate of $\mathcal{L}^*$ obtained in the first-stage, $\psi$ is defined in Strategy 2, and*

$$\hat{h}_{\mathrm{NMP}} \in \arg\min_{\mathcal{H}} \mathbb{P}_n \ell(h; z) - 2\psi \left( 16 \mathbb{P}_n \left[ (\ell(h; z) - \widehat{\mathcal{L}_0^*})^2 \right] \right).$$

*For any fixed $\delta \in (0, 1)$, by performing the moment-penalized estimator in Strategy 2, with probability at least $1 - \delta$,*

$$\mathscr{E}(\hat{h}_{\mathrm{MP}}) \leq 4\psi \left( 16\widehat{\mathcal{V}^*}; \frac{\delta}{C_n} \right) \vee \frac{r^*}{8B}, \tag{A.3}$$

*where $r^*$ is the fixed point of $16B\psi(r; \frac{\delta}{C_n})$.*

**Remarks.** 1) The subscript "NMP" within $\hat{h}_{\text{NMP}}$ means "negative moment penalization." Note that $\hat{h}_{\text{NMP}}$ may not have good generalization performance, it is only used to compute $\widehat{\mathcal{V}^*}$ so that we can evaluate the estimator $\hat{h}_{\text{MP}}$ proposed in Strategy 2.

2) While the fully data-dependent generalization error bound (A.3) provides a way to evaluate the moment-penalized estimator in Strategy 2 from training data, it seems that $\widehat{\mathcal{V}^*}$ and $\mathcal{V}^*$ are not necessarily of the same order. Therefore, (A.3) may not be as tight as the original variance-dependent rate in Theorem 2. One should view (A.3) as a relaxation of the original variance-dependent rate in Theorem 2.

3) We also comment that the "sub-root" assumption in Theorem 2 is not needed here as we do not discuss the precision of $\widehat{\mathcal{L}_0^*}$. It is easy to combine Corollary 4 with the guarantee on $\widehat{\mathcal{L}_0^*}$ proved in Appendix E.2.

The proof of Corollary 4 can be found in Appendix G.

# B    Application areas of problem-dependent rates

In Section 6 we illustrate the advantages of our problem-dependent rates in "rich" non-parametric and VC classes, where we use metric entropy conditions of the loss/excess loss class. In practical applications it is more standard to consider metric entropy conditions of the hypothesis class $\mathcal{H}$. In view of this, we introduce two important settings where the metric entropy on the loss/excess loss class can be obtained from metric entropy conditions on the hypothesis class $\mathcal{H}$. Thus, the improvements illustrated in Section 6 can be directly transferred to these application areas.

**Supervised learning with Lipchitz continuous cost.**    In supervised learning, the data $z$ is a feature-label pair $(x, y)$, and the loss $\ell(h; z)$ is of the form

$$\ell(h; z) = \ell_{\text{sv}}(h(x), y),$$

where $\ell_{\text{sv}}$ is a fixed cost function that is $L_{\text{sv}}-$Lipchitz continuous with respect to its first argument, namely, Lipchitz with parameter $L_{\text{sv}}$. For hypothesis classes characterized by metric entropy conditions, properties are preserved because

$$\log \mathcal{N}(\varepsilon, \ell \circ \mathcal{H}, L_2(\mathbb{P}_n)) \leq \log \mathcal{N}(\frac{\varepsilon}{L_{\text{sv}}}, \mathcal{H}, L_2(\mathbb{P}_n)).$$

Note that $L_{\text{sv}}$ only depends on the cost function and is usually of constant order. Our theory naturally applies to supervised learning problems where the cost function is Lipchitz continuous and not strongly-convex (for example, the $\ell_1$ cost, the hinge cost, the ramp cost, etc.).

**Counterfactual risk minimization.**    Denote $x \in \mathcal{X}$ the feature and $t \in \mathcal{T}$ the treatment (e.g. $\mathcal{T} = \{0, 1\}$ in binary treatment experimental design), and $c(x, t)$ the unknown cost function. A hypothesis (policy) $h$ is a map from $\mathcal{X} \times \mathcal{T}$ to $[0, 1]$ such that $\sum_{t \in \mathcal{T}} h(x, t) = 1$. Thus, a hypothesis (policy) essentially maps features to a distribution over treatments. We consider the standard formulation of "learning with logged bandit feedback," dubbed "counterfactual risk minimization" [15]: a batch of samples $\{(x_i, t_i, c_i)\}_{i=1}^n$ are obtained by applying a known policy $h_0$, so that $t_i$ is sampled from $h_0(x_i, \cdot)$ and one can only observe the cost $c_i$ associated with $t_i$. We write $z = (x, t, c)$ and let

$$\ell(h; z_i) = \frac{c_i}{h_0(x_i, t_i)} h(x_i, t_i), \tag{B.1}$$

be the "constructed loss" using importance sampling. It is straightforward to show that the population risk $\mathbb{P}\ell(h; z)$ is equal to the expected cost of policy $h$, so determining good policies requires one to minimize the generalization error $\mathscr{E}(\hat{h})$. It is usually convenient to obtain metric entropy condition of the loss/excess loss class by using the linearity structure of (B.1). In particular, from the Cauchy-Schwartz inequality we can prove that

$$\log \mathcal{N}(\varepsilon, \ell \circ \mathcal{H}, L_2(\mathbb{P}_n)) \leq \log \mathcal{N}(\frac{\varepsilon}{\gamma_n}, \mathcal{H}, L_4(\mathbb{P}_n)), \tag{B.2}$$

where $\gamma_n := \sqrt[4]{\mathbb{P}_n \left[ (\frac{c(x,t)}{h_0(x,t)})^4 \right]}$ only depends on the functions $c$, $h_0$ in the given problem, and the samples rather than the worst-case parameters. A systematical challenge in counterfactual risk

minimization is that the worst-case boundedness parameter, $\sup_{h,z} |\ell(h;z)|$, is typically very large, since the inverse probability term $\frac{1}{h_0(x_i,t_i)}$ in (B.1) is typically large in the worst case.

## C  Proof of Proposition 1

Given any $r_0 \in (0, R]$, take $r_k = 2^k r_0$, $k = 1, \cdots, \lceil \log_2 \frac{R}{r_0} \rceil$. Note that $\lceil \log_2 \frac{R}{r_0} \rceil \leq \log_2 \frac{2R}{r_0}$.

We use a union bound to establish that $\sup_{T(f) \leq r}(\mathbb{P} - \mathbb{P}_n)f \leq \psi(r; \delta)$ holds for all these $r_k$ simultaneously: $\forall \delta \in (0, 1)$, with probability at least $1 - \delta$,

$$\sup_{T(f) \leq r_k} (\mathbb{P} - \mathbb{P}_n)f \leq \psi\left(r_k; \frac{\delta}{\log_2 \frac{2R}{r_0}}\right), \quad k = 1, \cdots, \left\lceil \log_2 \frac{R}{r_0} \right\rceil.$$

For any fixed $f \in \mathcal{F}$, if $T(f) \leq r_0$ is false, then let $k$ be the non-negative integer such that $2^k r_0 < T(f) \leq 2^{k+1} r_0$, we further know that $r_{k+1} = 2^{k+1} r_0 \leq 2T(f)$. Therefore, with probability at least $1 - \delta$,

$$(\mathbb{P} - \mathbb{P}_n)f \leq \sup_{\tilde{f} \in \mathcal{F}: T(\tilde{f}) \leq r_{k+1}} (\mathbb{P} - \mathbb{P}_n)\tilde{f}$$

$$\leq \psi\left(r_{k+1}; \frac{\delta}{\log_2 \frac{2R}{r_0}}\right)$$

$$\leq \psi\left(2T(f); \frac{\delta}{\log_2 \frac{2R}{r_0}}\right).$$

Therefore, with probability at least $1 - \delta$, $\forall f \in \mathcal{F}$, either $T(f) \leq r_0$ or

$$(\mathbb{P} - \mathbb{P}_n)f \leq \psi\left(2T(f); \frac{\delta}{\log_2 \frac{2R}{r_0}}\right).$$

This completes the proof. □

## D  Proof of Theorem 1

Let $\mathcal{F}$ be the excess loss class in (3.2). Clearly, its members $f$ are uniformly bounded in $[-2B, 2B]$. Let $T(f) = \mathbb{P}[f^2]$. Define $\hat{f}$ by $\hat{f}(z) = \ell(\hat{h}_{\mathrm{ERM}}; z) - \ell(h^*; z), \forall z \in \mathcal{Z}$.

For a fixed $r_0 \in (0, 4B^2)$, Denote $C_{r_0} = 2 \log_2 \frac{8B^2}{r_0}$. Then from Proposition 1 we know with probability at least $1 - \frac{\delta}{2}$, either $T(\hat{f}) \leq r_0$ or

$$(\mathbb{P} - \mathbb{P}_n)\hat{f} \leq \psi\left(2T(\hat{f}); \frac{\frac{\delta}{2}}{\log_2 \frac{8B^2}{r_0}}\right) = \psi\left(2T(\hat{f}); \frac{\delta}{C_{r_0}}\right). \tag{D.1}$$

We denote the events $\mathscr{A}_1 = \{T(\hat{f}) \leq r_0\}$ and $\mathscr{A}_2 = \{$inequality (D.1) holds true$\}$, then we have

$$\mathrm{Prob}(\mathscr{A}_1) + \mathrm{Prob}(\mathscr{A}_1) \geq 1 - \frac{\delta}{2}.$$

Consider the event $\mathscr{A}_1$. From the surrogate property of $\psi$, we have

$$\mathrm{Prob}\left(\mathscr{A}_1 \cap \left\{(\mathbb{P} - \mathbb{P}_n)\hat{f} \leq \psi(2r_0; \frac{\delta}{C_{r_0}})\right\}\right) \geq \mathrm{Prob}(\mathscr{A}_1) - \frac{\delta}{C_{r_0}} \geq \mathrm{Prob}(\mathscr{A}_1) - \frac{\delta}{2}.$$

Combine the events $\mathscr{A}_1$ and $\mathscr{A}_2$, we have

$$\mathrm{Prob}\left(\left\{(\mathbb{P} - \mathbb{P}_n)\hat{f} \leq \psi\left(2T(\hat{f}) \vee 2r_0; \frac{\delta}{C_{r_0}}\right)\right\}\right) \geq \mathrm{Prob}(\mathscr{A}_1) - \frac{\delta}{2} + \mathrm{Prob}(\mathscr{A}_2) \geq 1 - \delta.$$

From the property of ERM we have $\mathbb{P}_n \hat{f} \leq 0$, so with probability at least $1 - \delta$,

$$\mathscr{E}(\hat{h}_{\mathrm{ERM}}) \leq (\mathbb{P} - \mathbb{P}_n)\hat{f} \leq \psi\left(2T(\hat{f}) \vee 2r_0; \frac{\delta}{C_{r_0}}\right). \tag{D.2}$$

From now to the end of this proof, we will prove the generalization error bound on the event

$$\mathscr{A} = \{\text{the inequality (D.2) holds true}\}, \tag{D.3}$$

whose measure is at least $1 - \delta$. Define $\hat{g}$ by $\hat{g}(z) = \ell(\hat{h}_{\mathrm{ERM}}; z) - \inf_{\mathcal{H}} \ell(h; z), \forall z \in \mathcal{Z}$. Let $T(\hat{g}) = \mathbb{P}[\hat{g}^2]$. We have $\hat{f}(z) = \hat{g}(z) - (\ell(h^*; z) - \inf_{\mathcal{H}} \ell(h; z)), \forall z$ so that

$$\mathbb{P}[\hat{f}^2] \leq 2\mathbb{P}[\hat{g}^2] + 2\mathbb{P}[(\ell(h^*; z) - \inf_{\mathcal{H}} \ell(h; z))^2]$$
$$\leq 2\mathbb{P}[\hat{g}^2] + 4B\mathcal{L}^* \leq 4\mathbb{P}[g^2] \vee 8B\mathcal{L}^*.$$

That is,

$$T(\hat{f}) \leq 4T(\hat{g}) \vee 8B\mathcal{L}^*. \tag{D.4}$$

From (D.2) and (D.4) we have

$$\mathbb{P}\hat{g} - \mathcal{L}^* = \mathscr{E}(\hat{h}_{\mathrm{ERM}}) \leq \psi\left(8T(\hat{g}) \vee 16B\mathcal{L}^* \vee 2r_0; \frac{\delta}{C_{r_0}}\right). \tag{D.5}$$

Since $\hat{g}(z) \in [0, 2B]$ for all $z$, we have $T(\hat{g}) \leq 2B\mathbb{P}\hat{g}$. From this fact and (D.5) we obtain

$$T(\hat{g}) \leq 2B\mathbb{P}\hat{g}$$
$$\leq 2B\left(\mathcal{L}^* + \psi\left(8T(\hat{g}) \vee 16B\mathcal{L}^* \vee 2r_0; \frac{\delta}{C_{r_0}}\right)\right)$$
$$= 2B\mathcal{L}^* + 2B\psi\left(8T(\hat{g}) \vee 16B\mathcal{L}^* \vee 2r_0; \frac{\delta}{C_{r_0}}\right).$$

Whether $B\mathcal{L}^*$ is less than $2B\psi\left(8T(\hat{g}) \vee 16B\mathcal{L}^* \vee 2r_0; \frac{\delta}{C_{r_0}}\right)$, or $B\mathcal{L}^*$ is greater or equal to $2B\psi\left(8T(\hat{g}) \vee 16B\mathcal{L}^* \vee 2r_0; \frac{\delta}{C_{r_0}}\right)$, the above inequality always implies that

$$T(\hat{g}) \leq 3B\mathcal{L}^* \vee 6B\psi\left(8T(\hat{g}) \vee 16B\mathcal{L}^* \vee 2r_0; \frac{\delta}{C_{r_0}}\right)$$
$$\leq 3B\mathcal{L}^* \vee 6B\psi\left(8T(\hat{g}); \frac{\delta}{C_{r_0}}\right) \vee 6B\psi\left(16B\mathcal{L}^* \vee 2r_0; \frac{\delta}{C_{r_0}}\right). \tag{D.6}$$

Let $r^*$ be the fixed point of $6B\psi(8r; \frac{\delta}{C_n})$. From the definition of fixed points whether $2B\mathcal{L}^* \vee \frac{r_0}{4} \leq r^*$ or $2B\mathcal{L}^* \vee \frac{r_0}{4} > r^*$, we always have

$$6B\psi\left(16B\mathcal{L}^* \vee 2r_0; \frac{\delta}{C_{r_0}}\right) \leq r^* \vee 2B\mathcal{L}^* \vee \frac{r_0}{4}.$$

Combine the above inequality with (D.6), we have

$$T(\hat{g}) \leq 3B\mathcal{L}^* \vee 6B\psi\left(8T(\hat{g}); \frac{\delta}{C_{r_0}}\right) \vee r^* \vee \frac{r_0}{4}.$$

From the above inequality and again the definition of fixed points, it is straightforward to prove that

$$T(\hat{g}) \leq 3B\mathcal{L}^* \vee r^* \vee \frac{r_0}{4}.$$

Combining the above inequality with (D.4), we have

$$T(\hat{f}) \leq 12B\mathcal{L}^* \vee 4r^* \vee r_0.$$

From the above inequality and (D.2) we have

$$\mathscr{E}(\hat{h}_{\mathrm{ERM}}) \le (\mathbb{P} - \mathbb{P}_n)\hat{f} \le \psi\left(24B\mathcal{L}^* \vee 8r^* \vee 2r_0; \frac{\delta}{C_{r_0}}\right), \tag{D.7}$$

which implies that

$$\mathscr{E}(\hat{h}_{\mathrm{ERM}}) \le \psi\left(24B\mathcal{L}^*; \frac{\delta}{C_{r_0}}\right) \vee \psi\left(8r^* \vee 2r_0; \frac{\delta}{C_{r_0}}\right).$$

Recall that $r^*$ is the fixed point of $6B\psi(8r; \frac{\delta}{C_{r_0}})$. Since $r^* \vee \frac{r_0}{4} \ge r^*$, from the definition of fixed points we have

$$6B\psi(8r^* \vee 2r_0; \frac{\delta}{C_{r_0}}) \le r^* \vee \frac{r_0}{4}.$$

So we finally obtain

$$\mathscr{E}(\hat{h}_{\mathrm{ERM}}) \le \psi\left(24B\mathcal{L}^*; \frac{\delta}{C_{r_0}}\right) \vee \frac{r^*}{6B} \vee \frac{r_0}{24B}.$$

Recall that the generalization error bound holds true on the event $\mathscr{A}$ defined in (D.3), whose measure is at least $1 - \delta$. This completes the proof. $\qquad\square$

# E    Proof of Theorem 2

The main goal of this subsection is to prove Theorem 2. We first prove Theorem 5 (the bound (5.2) in the main paper), a guarantee for the second-stage moment penalized estimator $\hat{h}_{\mathrm{MP}}$. In order to prove Theorem 2, we then combine Theorem 5 with a guarantee for the first-stage empirical risk minimization (ERM) estimator.

## E.1    Analysis for the second-stage moment-penalized estimator

**Theorem 5** (**variance-dependent rate of the second-stage estimator**). *Given arbitrary preliminary estimate $\mathcal{L}_{S'}^* \in [-B, B]$, the generalization error of the moment-penalized estimator $\hat{h}_{\mathrm{MP}}$ in Strategy 2 is bounded by*

$$\mathscr{E}(\hat{h}_{\mathrm{MP}}) \le 2\psi\left(c_0\left[\mathcal{V}^* \vee (\mathcal{L}_{S'}^* - \mathcal{L}_0^*)^2 \vee r^*\right]; \frac{\delta}{C_n}\right),$$

*with probability at least $1 - \delta$, where $c_0$ is an absolute constant and $r^*$ is the fixed point of $16B\psi(r; \frac{\delta}{C_n})$.*

**Proof of Theorem 5:**    the proof of Theorem 5 consist of four parts.

**Part I: use $\psi$ to upper bound localized empirical processes**
**Lemma 1** (**bound on localized empirical processes**). *Given a fixed $\delta_1 \in (0, 1)$, let $r_1^*(\delta_1)$ be the fixed point of $16B\psi(r; \delta_1)$ where $\psi$ is defined in Strategy 2. Then with probability at least $1 - \delta_1$, for all $r > 0$,*

$$\sup_{\mathbb{P}[f^2] \le r} (\mathbb{P} - \mathbb{P}_n)f \le \psi\left(r \vee r_1^*(\delta_1); \delta_1\right). \tag{E.1}$$

**Proof of Lemma 1:**    Recall that $\mathcal{F}$ is the excess loss class in (3.2). Clearly, its members $f$ are uniformly bounded in $[-2B, 2B]$. When $\mathbb{P}[f^2] \le r$, we have $\mathbb{P}[f^4] \le 4B^2 r$. From Lemma 4 (the two-sided version of its second inequality), with probability at least $1 - \frac{\delta_1}{2}$,

$$\sup_{\mathbb{P}[f^2] \le r} \left|(\mathbb{P} - \mathbb{P}_n)f^2\right|$$

$$\le 4\mathfrak{R}_n\{f^2 : \mathbb{P}[f^2] \le r\} + 2B\sqrt{\frac{2r\log\frac{8}{\delta_1}}{n}} + \frac{18B^2\log\frac{8}{\delta_1}}{n}$$

$$\le 16B\mathfrak{R}_n\{f : \mathbb{P}[f^2] \le r\} + 2B\sqrt{\frac{2r\log\frac{8}{\delta_1}}{n}} + \frac{18B^2\log\frac{8}{\delta_1}}{n},$$

where the last inequality follows from the Lipchitz contraction property of Rademahcer complexity (see, e.g., [10, Theorem 7]), and the fact that for all $f_1, f_2 \in \mathcal{F}$, $|f_1^2(z) - f_2^2(z)| \leq 4B|f_1(z) - f_2(z)|$. We conclude that with probability at least $1 - \frac{\delta_1}{2}$,

$$\sup_{\mathbb{P}[f^2] \leq r} \left| (\mathbb{P} - \mathbb{P}_n)f^2 \right| \leq \varphi_{\delta_1}(r), \tag{E.2}$$

where $\varphi_{\delta_1}(r) := 16B\mathfrak{R}_n\{f : \mathbb{P}[f^2] \leq r\} + 2B\sqrt{\frac{2r \log \frac{8}{\delta_1}}{n}} + \frac{18B^2 \log \frac{8}{\delta_1}}{n}$.

Denote $r_2^*(\delta_1)$ the fixed point of $4\varphi_{\delta_1}(r)$ (the fixed point must exist as $4\varphi_{\delta_1}(r)$ is a non-decreasing, non-negative and bounded function). From (E.2) and the fact that $r_2^*(\delta_1)$ is the fixed point of $4\varphi_{\delta_1}(r)$, if $r > r_2^*(\delta_1)$, then with probability at least $1 - \frac{\delta_1}{2}$,

$$\sup_{\mathbb{P}[f^2] \leq r} \left| (\mathbb{P} - \mathbb{P}_n)f^2 \right| \leq \frac{r}{4}. \tag{E.3}$$

(E.3) implies that with probability at least $1 - \frac{\delta_1}{2}$, for all $r > r_2^*(\delta_1)$, $\mathbb{P}[f^2] \leq r$ implies that

$$\mathbb{P}_n[f^2] \leq \frac{5}{4}r \leq 2r. \tag{E.4}$$

Again from the two-sided version of the second inequality in Lemma 4, we know that with probability at least $1 - \frac{\delta_1}{2}$,

$$\sup_{\mathbb{P}[f^2] \leq r} |(\mathbb{P} - \mathbb{P}_n)f| \leq 4\mathfrak{R}_n\{f : \mathbb{P}[f^2] \leq r\} + \sqrt{\frac{2r \log \frac{8}{\delta_1}}{n}} + \frac{9B \log \frac{8}{\delta_1}}{n}.$$

Combining the above inequality and (E.4) using a union bound, we know that with probability at least $1 - \frac{\delta_1}{2} - \frac{\delta_1}{2} = 1 - \delta_1$, if $r > r_2^*(\delta_1)$, then

$$\sup_{\mathbb{P}[f^2] \leq r} (\mathbb{P} - \mathbb{P}_n)f \leq 4\mathfrak{R}_n\{f : \mathbb{P}[f^2] \leq r\} + \sqrt{\frac{2r \log \frac{8}{\delta_1}}{n}} + \frac{9B \log \frac{8}{\delta_1}}{n}$$

$$\leq 4\mathfrak{R}_n\{f : \mathbb{P}_n[f^2] \leq 2r\} + \sqrt{\frac{2r \log \frac{8}{\delta_1}}{n}} + \frac{9B \log \frac{8}{\delta_1}}{n}. \tag{E.5}$$

Recall that the $\psi$ function satisfies that $\forall r > 0$,

$$4\mathfrak{R}_n\{f : \mathbb{P}_n[f^2] \leq 2r\} + \sqrt{\frac{2r \log \frac{8}{\delta_1}}{n}} + \frac{9B \log \frac{8}{\delta_1}}{n} \leq \psi(r; \delta_1).$$

From this fact and (E.5), we see that with probability at least $1 - \delta_1$, for all $r > 0$,

$$\sup_{\mathbb{P}[f^2] \leq r} (\mathbb{P} - \mathbb{P}_n)f \leq \psi\left(r \vee r_2^*(\delta_1); \delta_1\right). \tag{E.6}$$

From (E.6), in order to prove the result (E.1) in Lemma 1, we only need to prove that

$$r_2^*(\delta_1) \leq r_1^*(\delta_1). \tag{E.7}$$

Assume this is not true, i.e. $r_2^*(\delta_1) > r_1^*(\delta_1)$. Since $r_1^*(\delta_1)$ is the fixed point of $16B\psi(r; \delta_1)$, from the definition of fixed points we have

$$r_2^*(\delta_1) > 16B\psi(r_2^*(\delta_1); \delta_1).$$

From the definitions of $\psi$ and $\varphi_{\delta_1}$, for all $r > r_1^*(\delta_1)$,

$$4\varphi_{\delta_1}(r) \leq 16B\psi(r; \delta_1).$$

From the above two inequalities and $r_2^*(\delta_1) > r_1^*(\delta_1)$, we have

$$r_2^*(\delta_1) > 16B\psi(r_2^*(\delta_1); \delta_1) \geq 4\varphi_{\delta_1}(r_2^*(\delta_1)). \tag{E.8}$$

From the fact that $r_2^*(\delta_1)$ is the fixed point of $4\varphi_{\delta_1}$, we have

$$4\varphi_{\delta_1}(r_2^*(\delta_1)) = r_2^*(\delta_1). \tag{E.9}$$

The above two inequalities (E.8) and (E.9) result in a contradiction. So the assumption $r_2^*(\delta_1) > r_1^*(\delta_1)$ is false. Therefore $r_2^*(\delta_1) \leq r_1^*(\delta_1)$, and this completes the proof of Lemma 1. $\qquad\square$

**Part II: a "uniform localized convergence" argument with data-dependent measurement.**

Based on Lemma 1, we will modify the proof of Proposition 1 to obtain a "uniform localized convergence" argument with the data-dependent "measurement" functional $\mathbb{P}_n[f^2]$.

**Lemma 2** (**a "uniform localized convergence" argument with the data-dependent "measurement" functional**). *Given a fixed $\delta_1 \in (0,1)$, let $r_1^*(\delta_1)$ be the fixed point of $16B\psi(r;\delta_1)$ where $\psi$ is defined in Strategy 2. Then with probability at least $1 - 2\left(\log_2 \frac{8B^2 \vee 2r_1^*(\delta_1)}{r_1^*(\delta_1)}\right)\delta_1$, for all $f \in \mathcal{F}$ either $\mathbb{P}[f^2] \leq r_1^*(\delta_1)$, or*

$$(\mathbb{P} - \mathbb{P}_n)f \leq \psi\left(4\mathbb{P}_n[f^2]; \delta_1\right). \tag{E.10}$$

**Proof of Lemma 2:** from the definition of $\psi$ and the fact that $r_1^*(\delta_1)$ is the fixed point of $16B\psi(r;\delta_1)$, we know that $r_1^*(\delta_1) \geq \frac{144B^2 \log \frac{8}{\delta_1}}{n} > 0$. Take $r_0 = r_1^*(\delta_1)$.

Take $R = 4B^2 \vee r_0$ to be a uniform upper bound for $\mathbb{P}f^2$, and take $r_k = 2^k r_0, k = 1, \cdots, \lceil \log_2 \frac{R}{r_0} \rceil$. Note that $\lceil \log_2 \frac{R}{r_0} \rceil \leq \log_2 \frac{2R}{r_0}$. We use the union bound to establish that $\sup_{\mathbb{P}[f^2] \leq r}(\mathbb{P} - \mathbb{P}_n)f \leq \psi(r;\delta_1)$ holds for all $\{r_k\}$ simultaneously: with probability at least $1 - \log_2 \frac{2R}{r_0}\delta_1$,

$$\sup_{\mathbb{P}[f^2] \leq r_k}(\mathbb{P} - \mathbb{P}_n)f \leq \psi(r_k;\delta_1), \quad k = 1, \cdots, \left\lceil \log_2 \frac{R}{r_0} \right\rceil.$$

For any fixed $f \in \mathcal{F}$, if $\mathbb{P}[f^2] \leq r_0$ is false, let $k$ be the non-negative integer such that $2^k r_0 < \mathbb{P}[g(h;z)^2] \leq 2^{k+1}r_0$. We further have that $r_{k+1} = 2^{k+1}r_0 \leq 2\mathbb{P}[f^2]$. Therefore, with probability at least $1 - \log_2 \frac{2R}{r_0}\delta_1$,

$$\mathbb{P}f \leq \mathbb{P}_n f + \sup_{\tilde{f} \in \mathcal{F}: \mathbb{P}[\tilde{f}^2] \leq r_{k+1}}(\mathbb{P} - \mathbb{P}_n)\tilde{f}$$
$$\leq \mathbb{P}_n f + \psi(r_{k+1};\delta_1) \tag{E.11}$$

By (E.2) we know that with probability at least $1 - \frac{\delta_1}{2}$,

$$\sup_{\mathbb{P}[f^2] \leq r}\left(\mathbb{P}[f^2] - \mathbb{P}_n[f^2]\right) \leq \frac{r}{4}$$

for all $r > r_0$ (here we have used the fact $r_0 = r_1^*(\delta_1) \geq r_2^*(\delta_1)$, which is the result (E.7) in the proof of Lemma 1). From the union bound, with probability at least $1 - (\log_2 \frac{2R}{r_0} + \frac{1}{2})\delta_1 \geq 1 - 2(\log_2 \frac{2R}{r_0})\delta_1$, the condition $r_{k+1} \geq \mathbb{P}[f^2] > r_k$ will imply

$$\mathbb{P}_n[f^2] \geq \mathbb{P}[f^2] - \frac{1}{4}r_{k+1} \geq \frac{1}{4}r_{k+1},$$

so

$$r_{k+1} \leq 4\mathbb{P}_n[f^2].$$

Combining this result with (E.11), we have that for all $f$ such that $T(f) > r_0$, with probability at least $1 - 2(\log_2 \frac{2R}{r_0})\delta_1$,

$$\mathbb{P}f \leq \mathbb{P}_n f + \psi(r_{k+1};\delta_1)$$
$$\leq \mathbb{P}_n f + \psi\left(4\mathbb{P}_n[f^2];\delta_1\right).$$

We conclude that with probability at least $1 - 2(\log_2 \frac{2R}{r_0})\delta_1$, for all $f \in \mathcal{F}$, either $\mathbb{P}[f^2] \leq r_1^*(\delta_1)$, or

$$(\mathbb{P} - \mathbb{P}_n)f \leq \psi\left(4\mathbb{P}_n[f^2];\delta_1\right).$$

This completes the proof of Lemma 2. $\qquad\qquad\square$

**Part III: specify the moment-penalized estimator and its error bound.** We specify the moment-penalized estimator to be

$$\hat{h}_{\text{MP}} = \arg\min_{\mathcal{H}} \left\{ \mathbb{P}_n \ell(h; z) + \psi\left( 16\mathbb{P}_n[(\ell(h; z) - \mathcal{L}_{S'}^*)^2]; \delta_1 \right) \right\}. \tag{E.12}$$

Define $\hat{f}$ by $\hat{f}(z) = \ell(\hat{h}_{\text{MP}}; z) - \ell(h^*; z), \forall z \in \mathcal{Z}$, We define the event

$$\mathscr{A}_1 = \{ \mathbb{P}[\hat{f}^2] \leq r_1^*(\delta_1) \},$$

and event

$$\mathscr{A}_2 = \{ \text{the inequality (E.10) holds true at } \hat{f} \}.$$

Lemma 2 has proven that

$$\text{Prob}(\mathscr{A}_1) + \text{Prob}(\mathscr{A}_2) \geq 1 - 2\left( \log_2 \frac{8B^2 \vee 2r_1^*(\delta_1)}{r_1^*(\delta_1)} \right) \delta_1.$$

Consider the event $\mathscr{A}_1$ where $\mathbb{P}[\hat{f}^2] \leq r_1^*(\delta_1)$ holds true. Due to the surrogate property of $\psi$,

$$\text{Prob}\left( \mathscr{A}_1 \cap \left\{ (\mathbb{P} - \mathbb{P}_n)\hat{f} \leq \psi(r_1^*(\delta_1); \delta_1) \right\} \right) \geq \text{Prob}(\mathscr{A}_1) - \delta_1.$$

Combining events $\mathscr{A}_1$ and $\mathscr{A}_2$, we conclude that with probability at least $1 - 2\left( \log_2 \frac{8B^2 \vee 2r_1^*(\delta_1)}{r_1^*(\delta_1)} + 1 \right)\delta_1$, we have

$$(\mathbb{P} - \mathbb{P}_n)\hat{f} \leq \psi\left( 4\mathbb{P}_n[\hat{f}^2] \vee r_1^*(\delta_1); \delta_1 \right). \tag{E.13}$$

Denote $w(h; z) = \ell(h; z) - \mathcal{L}_{S'}^*$. Then $\hat{f}(z) = w(\hat{h}_{\text{MP}}; z) - w(h^*; z), \forall z \in \mathcal{Z}$, and we have that

$$4\mathbb{P}_n[\hat{f}^2] \leq 8\mathbb{P}_n[w(\hat{h}_{\text{MP}}; z)^2] + 8\mathbb{P}_n[w(h^*; z)^2]$$
$$\leq 16\mathbb{P}_n[w(\hat{h}_{\text{MP}}; z)^2] \vee 16\mathbb{P}_n[w(h^*; z)^2].$$

From the above conclusion and (E.13) we obtain with probability at least $1 - 2\left( \log_2 \frac{8B^2 \vee 2r_1^*(\delta_1)}{r_1^*(\delta_1)} + 1 \right)\delta_1$,

$$\mathscr{E}(\hat{h}_{\text{MP}}) + \mathbb{P}_n \ell(h^*; z) \leq \mathbb{P}_n \ell(\hat{h}_{\text{MP}}; z) + \psi(4\mathbb{P}_n[\hat{f}^2] \vee r_1^*(\delta_1); \delta_1)$$

$$\leq \mathbb{P}_n(\hat{h}_{\text{MP}}; z) + \psi\left( 16\mathbb{P}_n[w(\hat{h}_{\text{MP}}; z)^2] \vee 16\mathbb{P}_n[w(h^*; z)^2] \vee r_1^*(\delta_1); \delta_1 \right)$$

$$\leq \mathbb{P}_n(\hat{h}_{\text{MP}}; z) + \psi\left( 16\mathbb{P}_n[w(\hat{h}_{\text{MP}}; z)^2]\delta_1 \right) + \psi\left( 16\mathbb{P}_n[w(h^*; z)^2] \vee r_1^*(\delta_1); \delta_1 \right). \tag{E.14}$$

From the definition (E.12) of $\hat{h}_{\text{MP}}$, we have

$$\mathbb{P}_n \ell(\hat{h}_{\text{MP}}; z) + \psi\left( 16\mathbb{P}_n[w(\hat{h}_{\text{MP}}; z)^2]; \delta_1 \right) \leq \mathbb{P}_n \ell(h^*; z) + \psi\left( 16\mathbb{P}_n[w(h^*; z)^2]; \delta_1 \right) \tag{E.15}$$

Therefore, with probability at least $1 - 2\left( \log_2 \frac{8B^2 \vee 2r_1^*(\delta_1)}{r_1^*(\delta_1)} + 1 \right)\delta_1$,

$$\mathscr{E}(\hat{h}_{\text{MP}}) \leq \mathbb{P}_n \ell(\hat{h}_{\text{MP}}; z) + \psi\left( 16\mathbb{P}_n[w(\hat{h}_{\text{MP}}; z)^2]; \delta_1 \right) + \psi\left( 16\mathbb{P}_n[w(h^*; z)^2] \vee r_1^*(\delta_1); \delta_1 \right) - \mathbb{P}_n \ell(h^*; z)$$

$$= \arg\min_{\mathcal{H}} \left\{ \mathbb{P}_n \ell(h; z) + \psi\left( 16\mathbb{P}_n[w(h; z)]; \delta_1 \right) \right\} - \mathbb{P}_n \ell(h^*; z) + \psi\left( 16\mathbb{P}_n[w(h^*; z)^2] \vee r_1^*(\delta_1); \delta_1 \right)$$

$$\leq \psi\left( 16\mathbb{P}_n[w(h^*; z)^2]; \delta_1 \right) + \psi\left( 16\mathbb{P}_n[w(h^*; z)^2] \vee r_1^*(\delta_1); \delta_1 \right)$$

$$\leq 2\psi\left( 16\mathbb{P}_n[w(h^*; z)^2] \vee r_1^*(\delta_1); \delta_1 \right), \tag{E.16}$$

where the first inequality is due to (E.14) and the second inequality is due to (E.15).

From Bernstein's inequality at the single element $h^*$, for any fixed $\delta_2 \in (0, 1)$, with probability at least $1 - \delta_2$,

$$\mathbb{P}_n[w(h^*; z)^2] \le \mathbb{P}[w(h^*; z)^2] + 2B\sqrt{\frac{2\mathbb{P}[w(h^*; z)^2]\log\frac{2}{\delta_2}}{n}} + \frac{4B^2 \log\frac{2}{\delta_2}}{n}$$

$$\le 2\mathbb{P}[w(h^*; z)^2] + \frac{6B^2 \log\frac{2}{\delta_2}}{n}.$$

Therefore, we conclude that with probability at least $1 - 2\left(\log_2 \frac{8B^2 \vee 2r_1^*(\delta_1)}{r_1^*(\delta_1)} + 1\right)\delta_1 - \delta_2$,

$$\mathscr{E}(\hat{h}_{\mathrm{MP}}) \le 2\psi\left(16\mathbb{P}_n[w(h^*; z)] \vee r_1^*(\delta_1) \vee \frac{B^2}{n}; \delta_1\right)$$

$$\le 2\psi\left(\left(32\mathbb{P}[w(h^*; z)^2] + \frac{96B^2 \log\frac{2}{\delta_2}}{n}\right) \vee r_1^*(\delta_1) \vee \frac{B^2}{n}; \delta_1\right). \qquad (E.17)$$

**Part IV: final steps.**

From the definition of $\psi$ and the fact that $r_1^*(\delta_1)$ is the fixed point of $16B\psi(r; \delta_1)$, we know that

$$r_1^*(\delta_1) \ge \frac{144B^2 \log\frac{8}{\delta_1}}{n}. \qquad (E.18)$$

Denote $C_n := 4\log_2 n + 10 \ge 4\log_2$ and take

$$\delta_1 = \frac{\delta}{C_n}, \qquad (E.19)$$

then we have

$$4\log_2 \frac{8B^2 \vee 2r_1^*(\delta_1)}{r_1^*(\delta_1)} + 6 \le \max\left\{4\log_2 \frac{8n}{144\log 8}, 4 + 6\right\}$$

$$\le \max\{4\log_2 n, 10\} \le C_n,$$

so

$$\left(2\log_2 \frac{8B^2 \vee 2r_1^*(\delta_3)}{r_1^*(\delta_3)} + 3\right)\delta_1 \le \frac{\delta}{2}.$$

Set $r^* = r_1^*(\delta_1)$ and take $\delta_2 = \frac{\delta}{2}$. From (E.17), we obtain that with probability at least $1 - \delta$, the generalization error of $\hat{h}_{\mathrm{MP}}$ is upper bounded by

$$\mathscr{E}(\hat{h}_{\mathrm{MP}}) \le 2\psi\left(c\left[\mathbb{P}[w(h^*; z)^2] \vee r^* \vee \frac{B^2 \log\frac{4}{\delta}}{n}\right]; \frac{\delta}{C_n}\right), \qquad (E.20)$$

where $c$ is an absolute constant. From (E.18) we have $r_1^*(\delta_1) \ge \frac{144B^2 \log\frac{8C_n}{\delta}}{n} \ge \frac{B^2 \log\frac{4}{\delta}}{n}$. Combine this fact with the inequality (E.20), we obtain that

$$\mathscr{E}(\hat{h}_{\mathrm{MP}}) \le 2\psi\left(c\left[\mathbb{P}[(\ell(h^*; z) - \mathcal{L}_{S'}^*)^2] \vee r^*\right]; \frac{\delta}{C_n}\right)$$

$$\le 2\psi\left(c_0\left[\mathcal{V}^* \vee r^* \vee (\mathcal{L}_{S'}^* - \mathcal{L}_0^*)^2\right]; \frac{\delta}{C_n}\right). \qquad (E.21)$$

where $c_0$ is an absolute constant. This completes the proof of Theorem 5. $\qquad\square$

### E.2 Analysis of the first-stage ERM estimator

After proving Theorem 5, the remaining part needed to prove Theorem 2 is to bound $(\mathcal{L}_{S'}^* - \mathcal{L}_0^*)^2$—the error of the first-stage ERM estimator.

**The remaining steps in the proof of Theorem 2:** We will give a guarantee on the first-stage ERM estimator, and combine this guarantee with Theorem 5 to prove Theorem 2. Recall that $\mathbb{P}_{S'}$ is the empirical distribution of the "auxiliary" data set. Denote $\hat{h}_{\text{ERM}} \in \arg\min_{\mathcal{H}} \mathbb{P}_{S'}\ell(h; z)$.

From Part I in the proof of Theorem 5, $\forall \delta \in (0, \frac{1}{2})$, with probability at least $1 - \delta$,

$$\sup_{\mathcal{F}} |(\mathbb{P} - \mathbb{P}_n)f| \leq \psi(4B^2; \delta) \leq \psi\left(4B^2; \frac{\delta}{C_n}\right).$$

Since $\psi$ is sub-root with respect to its first argument, we have

$$\frac{\psi(4B^2; \frac{\delta}{C_n})}{\sqrt{4B^2}} \leq \frac{\psi(r^*; \frac{\delta}{C_n})}{\sqrt{r^*}} = \frac{\sqrt{r^*}}{16B},$$

where $r^*$ is the fixed point of $16B\psi(r; \frac{\delta}{C_n})$. So we have proved that $\psi(4B^2; \frac{\delta}{C_n}) \leq \frac{\sqrt{r^*}}{8}$. Therefore,

$$\sup_{\mathcal{F}} |(\mathbb{P} - \mathbb{P}_n)f| \leq \frac{\sqrt{r^*}}{8}.$$

Because $\hat{h}_{\text{ERM}} \in \arg\min_{\mathcal{H}} \mathbb{P}_{S'}\ell(h; z)$ and $\mathbb{P}_{S'}\ell(\hat{h}_{\text{ERM}}; z) = \mathcal{L}_{S'}^*$, we have

$$
\begin{aligned}
\mathcal{L}_{S'}^* - \mathcal{L}_0^* &= (\mathbb{P}_{S'}\ell(\hat{h}_{\text{ERM}}; z) - \mathbb{P}_{S'}\ell(h^*; z)) + (\mathbb{P}_{S'}\ell(h^*; z) - \mathbb{P}\ell(h^*; z)) \\
&\leq \mathbb{P}_{S'}\ell(h^*; z) - \mathbb{P}\ell(h^*; z) \leq \sup_{\mathcal{F}} |(\mathbb{P} - \mathbb{P}_n)f|,
\end{aligned}
$$

and

$$
\begin{aligned}
\mathcal{L}_{S'}^* - \mathcal{L}_0^* &= (\mathbb{P}_{S'}\ell(\hat{h}_{\text{ERM}}; z)) - \mathbb{P}\ell(\hat{h}_{\text{ERM}}; z)) + (\mathbb{P}\ell(\hat{h}_{\text{ERM}}; z) - \mathbb{P}\ell(h^*; z)) \\
&\geq \mathbb{P}_{S'}\ell(\hat{h}_{\text{ERM}}; z)) - \mathbb{P}\ell(\hat{h}_{\text{ERM}}; z) \geq - \sup_{\mathcal{F}} |(\mathbb{P} - \mathbb{P}_n)f|.
\end{aligned}
$$

Hence we have

$$(\mathcal{L}_{S'}^* - \mathcal{L}_0^*)^2 \leq (\sup_{\mathcal{F}} |(\mathbb{P} - \mathbb{P}_n)f|)^2 \leq \frac{r^*}{64}.$$

Combine this result with (E.21), we have that $\forall \delta \in (0, \frac{1}{2})$, with probability $1 - 2\delta$,

$$
\begin{aligned}
\mathscr{E}(\hat{h}_{\text{MP}}) &\leq 2\psi\left(c_1 \left(\mathcal{V}^* \vee r^*\right); \frac{\delta}{C_n}\right) \\
&\leq 2\left(\psi\left(c_1\mathcal{V}^*; \frac{\delta}{C_n}\right) \vee \psi\left(c_1 r^*; \frac{\delta}{C_n}\right)\right) \\
&\leq 2\psi\left(c_1\mathcal{V}^*; \frac{\delta}{C_n}\right) \vee \frac{c_1 r^*}{8B},
\end{aligned}
$$

where $c_1 = \max\{c_0, 16\}$ is an absolute constant, and the last inequality follows from the fact that $\frac{c_1 r^*}{16} > r^*$ and the definition of fixed points. This completes the proof of Theorem 2. $\qquad\square$

# F   Proof of Corollary 3

From the definitions, we know that $\mathcal{L}^* = \mathbb{P}[\ell(h^*; z) - \inf_{\mathcal{H}} \ell(h^*; z)]$, $\widehat{\mathcal{L}^*} = \mathbb{P}_n[\ell(\hat{h}_{\text{ERM}}; z) - \inf_{\mathcal{H}} \ell(h; z)]$ and $\mathbb{P}\ell(h^*; z) \leq \mathbb{P}\ell(\hat{h}_{\text{ERM}}; z)$. As a result, we have

$$
\begin{aligned}
\mathcal{L}^* - \widehat{\mathcal{L}^*} &= \mathbb{P}\ell(h^*; z) - \mathbb{P}_n\ell(\hat{h}_{\text{ERM}}; z) - (\mathbb{P} - \mathbb{P}_n)[\inf_{\mathcal{H}} \ell(h; z)] \\
&\leq (\mathbb{P} - \mathbb{P}_n)\ell(\hat{h}_{\text{ERM}}; z) - (\mathbb{P} - \mathbb{P}_n)[\inf_{\mathcal{H}} \ell(h; z)] \\
&= (\mathbb{P} - \mathbb{P}_n)\hat{f} + (\mathbb{P} - \mathbb{P}_n)[\ell(h^*; z) - \inf_{\mathcal{H}} \ell(h; z)], \quad\quad\quad\text{(F.1)}
\end{aligned}
$$

where $\hat{f}$ is defined by $\hat{f}(z) = \ell(\hat{h}_{\text{ERM}}; z) - \ell(h^*; z), \forall z \in \mathcal{Z}$.

We take $r_0 = \frac{B^2}{n}$ in Theorem 1, and denote $C_n := C_{r_0} = 2\log_2 n + 6$. From (D.7) in the proof of Theorem 1, on the event $\mathscr{A}$ defined in (D.3) (whose measure is at least $1 - \delta$),

$$\mathscr{E}(\hat{h}_{\mathrm{ERM}}) \leq (\mathbb{P} - \mathbb{P}_n)\hat{f} \leq \psi(24B\mathcal{L}^* \vee 8r^* \vee \frac{2B^2}{n}; \frac{\delta}{C_n}). \tag{F.2}$$

Since $3B\mathcal{L}^* \vee r^* \vee \frac{B^2}{4n} \geq r^*$, from the definition of fixed points we have

$$(\mathbb{P} - \mathbb{P}_n)\hat{f} \leq \psi\left(8\left(3B\mathcal{L}^* \vee r^* \vee \frac{B^2}{4n}\right); \frac{\delta}{C_n}\right)$$

$$\leq \frac{3B\mathcal{L}^* \vee r^* \vee \frac{B^2}{4n}}{6B} \leq \frac{\mathcal{L}^*}{2} + \frac{r^*}{6B} + \frac{B}{24n}. \tag{F.3}$$

This result holds together with the result of Theorem 1 on the event $\mathscr{A}$.

The random variable $\ell(h^*; z) - \inf_{\mathcal{H}} \ell(h; z)$ is uniformly bounded by $[0, 2B]$. From Bernstein's inequality and the fact $\mathrm{Var}[\ell(h^*; z) - \inf_{\mathcal{H}} \ell(h; z)] \leq 2B\mathcal{L}^*$, with probability at least $1 - \delta$,

$$\left|(\mathbb{P} - \mathbb{P}_n)[\ell(h^*; z) - \inf_{\mathcal{H}} \ell(h; z)]\right| \leq \sqrt{\frac{4B\mathcal{L}^* \log\frac{2}{\delta}}{n}} + \frac{2B\log\frac{2}{\delta}}{n} \leq \frac{\mathcal{L}^*}{4} + \frac{3B\log\frac{2}{\delta}}{n}. \tag{F.4}$$

Consider the event

$$\mathscr{A}_3 = \mathscr{A} \cup \{\text{inequality (F.4) holds true}\},$$

whose measure is at least $1 - 2\delta$. On the event $\mathscr{A}_3$, from inequalities (F.1) (F.3) (F.4), it is straightforward to show that

$$\mathcal{L}^* - \widehat{\mathcal{L}^*} \leq \frac{3}{4}\mathcal{L}^* + \frac{r^*}{6B} + \frac{4B\log\frac{2}{\delta}}{n},$$

which implies

$$\mathcal{L}^* \leq 4\widehat{\mathcal{L}^*} + \frac{2r^*}{3B} + \frac{16B\log\frac{2}{\delta}}{n}. \tag{F.5}$$

From this result and (F.2), it is straightforward to show that

$$\mathscr{E}(\hat{h}_{\mathrm{ERM}}) \leq \psi\left(cB\widehat{\mathcal{L}^*}; \frac{\delta}{C_n}\right) \vee \frac{cr^*}{n} \vee \frac{cB\log\frac{2}{\delta}}{n},$$

where $c$ is an absolute constant.

We also have

$$\widehat{\mathcal{L}^*} - \mathcal{L}^* = \mathbb{P}_n\ell(\hat{h}_{\mathrm{ERM}}) - \mathbb{P}\ell(h^*; z) - (\mathbb{P}_n - \mathbb{P})[\inf_{\mathcal{H}} \ell(h; z)]$$

$$\leq (\mathbb{P}_n - \mathbb{P})\ell(h^*; z) - (\mathbb{P}_n - \mathbb{P})[\inf_{\mathcal{H}} \ell(h; z)]$$

$$= (\mathbb{P}_n - \mathbb{P})[\ell(h^*; z) - \inf_{\mathcal{H}} \ell(h; z)].$$

From this result and (F.4), on the event $\mathscr{A}_3$,

$$\widehat{\mathcal{L}^*} \leq \frac{5}{4}\mathcal{L}^* + \frac{3B\log\frac{2}{\delta}}{n}. \tag{F.6}$$

Combine (F.5) and (F.6) we obtain

$$\mathcal{L}^* \leq c_1\left(\widehat{\mathcal{L}^*} \vee \frac{r^*}{B} \vee \frac{B\log\frac{2}{\delta}}{n}\right) \leq c_2\left(\mathcal{L}^* \vee \frac{r^*}{B} \vee \frac{B\log\frac{2}{\delta}}{n}\right),$$

where $c_1$ and $c_2$ are absolute constants. This completes the proof. $\qquad\square$

# G  Proof of Corollary 4

Define $\hat{f}_{\mathrm{NMP}}$ by $\hat{f}_{\mathrm{NMP}}(z) = \ell(\hat{h}_{\mathrm{NMP}}; z) - \ell(h^*; z), \forall z \in \mathcal{Z}$, and $w(h; z) = \ell(h; z) - \widehat{\mathcal{L}_0^*}$. In the proof of Theorem 5, the result (E.16) and the specification of $\delta_1$ in (E.19) show that with probability at least $1 - \frac{\delta}{2}$,

$$\mathscr{E}(\hat{h}_{\mathrm{MP}}) \leq 2\psi\left(16\mathbb{P}_n[w(h^*; z)^2] \vee r^*; \frac{\delta}{C_n}\right). \tag{G.1}$$

We also refer to another implication of the proof of Theorem 5. Note that the proof of the result (E.13) does not depends on any property of the estimator $\hat{h}_{\mathrm{MP}}$. By the repeating the lines between (E.12) and (E.13) for the estimator $\hat{h}_{\mathrm{NMP}}$, and use the specification of $\delta_1$ in (E.19), it is straightforward to show that with probability at least $1 - \frac{\delta}{2}$,

$$(\mathbb{P} - \mathbb{P}_n)\hat{f}_{\mathrm{NMP}} \leq \psi\left(4\mathbb{P}_n[\hat{f}_{\mathrm{NMP}}^2] \vee r^*; \frac{\delta}{C_n}\right). \tag{G.2}$$

We continue the proof on the event

$$\mathscr{A} := \{\text{the inequalities (G.2) and (G.1) hold true}\},$$

whose measure is at least $1 - \delta$.

From the definition of $\hat{h}_{\mathrm{NMP}}$,

$$\mathbb{P}_n\ell(\hat{h}_{\mathrm{NMP}}; z) - 2\psi\left(16\mathbb{P}_n[w(\hat{h}_{\mathrm{NMP}}; z)^2]; \frac{\delta}{C_n}\right) \leq \mathbb{P}_n\ell(h^*; z) - 2\psi\left(16\mathbb{P}_n[w(h^*; z)^2]; \frac{\delta}{C_n}\right). \tag{G.3}$$

Therefore, we have

$$2\psi\left(16\mathbb{P}_n[w(h^*; z)^2]; \frac{\delta}{C_n}\right)$$

$$\leq 2\psi\left(16\mathbb{P}_n[w(\hat{h}_{\mathrm{NMP}}; z)^2]; \frac{\delta}{C_n}\right) + \mathbb{P}_n\ell(h^*; z) - \mathbb{P}_n\ell(\hat{h}_{\mathrm{NMP}}; z)$$

$$= 2\psi\left(16\mathbb{P}_n[w(\hat{h}_{\mathrm{NMP}}; z)^2]; \frac{\delta}{C_n}\right) + \mathbb{P}[\ell(h^*; z) - \ell(\hat{h}_{\mathrm{NMP}}; z)] + (\mathbb{P}_n - \mathbb{P})[\ell(h^*; z) - \ell(\hat{h}_{\mathrm{NMP}}; z)]$$

$$\leq 2\psi\left(16\mathbb{P}_n[w(\hat{h}_{\mathrm{NMP}}; z)^2]; \frac{\delta}{C_n}\right) + (\mathbb{P} - \mathbb{P}_n)\hat{f}_{\mathrm{NMP}}$$

$$\leq 2\psi\left(16\mathbb{P}_n[w(\hat{h}_{\mathrm{NMP}}; z)^2]; \frac{\delta}{C_n}\right) + \psi\left(4\mathbb{P}_n[\hat{f}_{\mathrm{NMP}}^2]; \frac{\delta}{C_n}\right), \tag{G.4}$$

where the first inequality is due to (G.3), the second inequality is due to the fact that $h^*$ minimizes the population risk; and the last inequality is due to (G.2).

Note that

$$4\mathbb{P}_n[\hat{f}_{\mathrm{NMP}}^2] \leq 8\mathbb{P}_n[w(\hat{h}_{\mathrm{NMP}}; z)^2] + 8\mathbb{P}_n[w(h^*; z)^2]$$

$$\leq 16\mathbb{P}_n[w(\hat{h}_{\mathrm{NMP}}; z)^2] \vee 16\mathbb{P}_n[w(h^*; z)^2].$$

From the above inequality and (G.4), we have

$$2\psi\left(16\mathbb{P}_n[w(h^*; z)^2]; \frac{\delta}{C_n}\right)$$

$$\leq 2\psi\left(16\mathbb{P}_n[w(\hat{h}_{\mathrm{NMP}}; z)^2]; \frac{\delta}{C_n}\right) + \psi\left(16\mathbb{P}_n[w(\hat{h}_{\mathrm{NMP}}; z)^2]; \frac{\delta}{C_n}\right) \vee \psi\left(16\mathbb{P}_n[w(h^*; z)^2]; \frac{\delta}{C_n}\right). \tag{G.5}$$

Whether $\mathbb{P}_n[w(h^*; z)^2] \leq 16\mathbb{P}_n[w(\hat{h}_{\mathrm{NMP}}; z)^2]$ or $\mathbb{P}_n[w(h^*; z)^2] > 16\mathbb{P}_n[w(\hat{h}_{\mathrm{NMP}}; z)^2]$, the inequality (G.5) always implies

$$\psi\left(16\mathbb{P}_n[w(h^*; z)^2]; \frac{\delta}{C_n}\right) \leq 2\psi\left(16\mathbb{P}_n[w(\hat{h}_{\mathrm{NMP}}; z)^2]; \frac{\delta}{C_n}\right) = 2\psi\left(16\widehat{\mathcal{V}^*}; \frac{\delta}{C_n}\right). \tag{G.6}$$

(Note that $\widehat{\mathcal{V}^*} := \mathbb{P}_n[w(\hat{h}_{\mathrm{NMP}}; z)^2].$) We conclude that with probability at least $1 - \delta$,

$$
\begin{aligned}
\mathscr{E}(\hat{h}_{\mathrm{MP}}) &\leq 2\psi\left(16\mathbb{P}_n[w(h^*; z)^2] \vee r^*; \frac{\delta}{C_n}\right) \\
&= 2\psi\left(16\mathbb{P}_n[w(h^*; z)^2]; \frac{\delta}{C_n}\right) \vee 2\psi(r^*; \frac{\delta}{C_n}) \\
&\leq 4\psi\left(16\widehat{\mathcal{V}^*}; \frac{\delta}{C_n}\right) \vee \frac{r^*}{8B},
\end{aligned}
$$

where the first inequality is due to (G.1) and the last inequality is due to (G.6). This completes the proof. $\qquad\square$

## H  Auxiliary lemmas

**Lemma 3 (Dudley's integral bound, [14]).** *Given $r > 0$ and a class $\mathcal{F}$ that consists of functions defined on $\mathcal{Z}$,*

$$
\mathfrak{R}_n\{f \in \mathcal{F} : \mathbb{P}_n[f^2] \leq r\} \leq \inf_{\varepsilon_0 > 0}\left\{4\varepsilon_0 + 12\int_{\varepsilon_0}^{\sqrt{r}}\sqrt{\frac{\log \mathcal{N}(\varepsilon, \mathcal{F}, L_2(\mathbb{P}_n))}{n}}d\varepsilon\right\}.
$$

**Lemma 4 (Talagrand's concentration inequality for empirical processes, [2]).** *Let $\mathcal{F}$ be a class of functions that map $\mathcal{Z}$ into $[B_1, B_2]$. Assume that there is some $r > 0$ such that for every $f \in \mathcal{F}$, $\mathrm{Var}[f(z_i)] \leq r$. Then, for every $\delta \in (0, 1)$, with probability at least $1 - \delta$,*

$$
\sup_{f \in \mathcal{F}}(\mathbb{P} - \mathbb{P}_n)f \leq 3\mathfrak{R}\mathcal{F} + \sqrt{\frac{2r\log\frac{1}{\delta}}{n}} + (B_2 - B_1)\frac{\log\frac{1}{\delta}}{n},
$$

*and with probability at least $1 - \delta$,*

$$
\sup_{f \in \mathcal{F}}(\mathbb{P} - \mathbb{P}_n)f \leq 4\mathfrak{R}_n\mathcal{F} + \sqrt{\frac{2r\log\frac{2}{\delta}}{n}} + \frac{9}{2}(B_2 - B_1)\frac{\log\frac{2}{\delta}}{n}.
$$

*Moreover, the same results hold for the quantity $\sup_{f \in \mathcal{F}}(\mathbb{P}_n - \mathbb{P})f$.*

**Lemma 5 (Bernstein's inequality).** *Let $X_1, \cdots, X_n$ be real-valued, independent, mean-zero random variables and suppose that for some constants $\sigma, B > 0$,*

$$
\frac{1}{n}\sum_{i=1}^n \mathbb{E}|X_i|^k \leq \frac{k!}{2}\sigma^2 B^{k-2}, \quad k = 2, 3, \cdots
$$

*Then, $\forall \delta \in (0, 1)$, with probability at least $1 - \delta$*

$$
\left|\frac{1}{n}\sum_{i=1}^n X_i\right| \leq \sqrt{\frac{2\sigma^2\log\frac{2}{\delta}}{n}} + \frac{B\log\frac{2}{\delta}}{n}. \tag{H.1}
$$