[Reviews · NeurIPS 2020]

Review 1

Summary and Contributions: The paper suggests a more subtle analysis of uniform localize convergence that is problem-dependent in the following sense: each problem is defined by its best hypothesis (not necessarily from the hypothesis set), and the problem-dependency bounds are either by the loss or the variance of this hypothesis. The paper shows that this approach yields a lower bound to the problem-independent analysis (like local Rademacher complexity), with equality only for a subset of the problems. Finally, the paper shows examples of implementing those bounds.

Strengths: The paper confronts the known problem-independent techniques which are known to be far from optimal in some problems, and suggest a new strategy that fits each problem differently. Each of the three examples in section 6 can be considered a specific contribution by itself, as each emphasis on how different distributions can lead to different bounds; thus, distribution-independent bounds are assumed to be far from optimal. The paper is well written and provides good intuition of the work and the proofs of the claims. After each statement or theorem, the authors give a brief explanation, that put their claims in context and actively helped for the readability.

Weaknesses: The dependency on L* and V* is theoretically acceptable; however it is unclear how one can estimate them in practice. The authors suggested to add a new corolarry for this case. There are some notations that were explained by the author during the rebuttle phase and should be added to the paper itself: Explain that psi is dependent of n, and the arithmetic definition of [\vee].

Correctness: While I did not thoroughly review the proofs, the proofs' intuitions do not raise any red flag.

Clarity: The paper is well written and organized.

Relation to Prior Work: The authors provide great emphasis to explain their contribution over previous works.

Reproducibility: Yes

Additional Feedback: Minor issues: - Line 86: you defined h in H, but used f in equation 2.5. - The paragraph after line 89: it is missing r_0 after upper bounding T(f).


Review 2

Summary and Contributions: This paper examines generalization errors that depend on the loss and variance of the best hypotheses. The analysis is based on refinements of local Rademacher complexity arguments. Additionally, the paper proposes a new moment-penalized estimator and examines the rates obtained for a few nonparametric estimators. The rates improve on recent work in various parameter regimes.

Strengths: The theoretical results are certainly of interest, given the community push to explain generalization of sophisticated estimators without relying on basic uniform convergence bounds. Although I didn't check the proofs carefully, it appears that an above-average amount of technical effort went into the results.

Weaknesses: 1) The push to better understand generalization is mostly driven by neural networks, so if the bounds could be described for neural networks, that would be helpful. Of course, this presumably requires a good bound on \psi. (2) It also may be helpful to include a more concrete discussion on why the resulting technique is unhelpful for parametric classes, where local Rademacher complexity gives optimal results. In general, it would help to make the comparisons clearer.

Correctness: The paper seems correct, although I did not check the proofs carefully.

Clarity: (3) The paper could be written better. I laughed a bit when I saw that Section 3 was the preliminaries. Usually section 2 is the preliminaries so that it doesn't break the flow of the main arguments of the paper. (4) In Section 6 (and elsewhere, where appropriate), I would have liked a table giving some explicit rate comparisons. Sure, there are discussions of the rate comparisons in the text, but they're simply not as easy to follow, especially when many of the aligned expressions aren't really equations or inequalities in the sense that they describe a relation but merely an expression. I find that this makes it harder to identify key information quickly. L134: characterizes L160: the left and right quotes are not in the same style L232: hypothesis L246: variance-dependent L264: invoking

Relation to Prior Work: Likely enough for someone in this area.

Reproducibility: Yes

Additional Feedback: I have read the other reviews and the response and have updated my score.


Review 3

Summary and Contributions: This paper proposes a new analysis framework, which is termed “uniform localized convergence”, that allows for a finer characterization of problem-dependent rates in learning. In particular, Theorem 1 characterizes the loss-dependent rate of ERM and Section 6.2 illustrates how this can lead to tighter rates than the traditional ones known. In addition, Theorem 2 proposes a new estimator with a better variance-dependent rate than what is known in prior work. Finally, Section 6 illustrates some examples where we get improved rates based on Theorems 1 and 2.

Strengths: I think this is a novel contribution with very clean and crisp results. Beyond proving improved problem-dependent rates, the machinery and techniques involved may be useful in other applications as well.

Weaknesses: For variance dependent rates, perhaps the authors could discuss a bit more on the tradeoffs between practicality of the proposed estimator and how that compares with prior work. This is lightly discussed in the broader impact section, but maybe this could be expanded on a bit further. Also, do we know whether the rates in Theorems 1 and 2 are optimal or not? Perhaps the authors could discuss cases where this is or (is not) the case.

Correctness: I haven’t spotted any issues with the claims/proofs.

Clarity: Very well-written and clear.

Relation to Prior Work: Yes.

Reproducibility: Yes

Additional Feedback: *** Update after author response *** Thank you for addressing my questions in the response.


Review 4

Summary and Contributions: The authors derive new, sharper bounds for the generalization error in the standard statistical learning framework. Their general results hold under weaker assumptions (not requiring the Bernstein condition and sub-root property of the \psi surrogate function). They derive both a loss and a variance dependent version of the upper bound and apply the derived results on nonparametric function classes demonstrating the advantages of their results over standard approaches in the literature.

Strengths: The paper is well written, I liked that the authors provided proof-sketches in the paper, not deferring everything technical to the supplement. They apply standard, but involved empirical process techniques, hence the level of the paper is higher than standard NeurIPS submissions. The derived results are clean and elegant and the structure of the paper is also good.

Weaknesses: Beside the many typos (see below for a non-complete list) in several places the derived results are not rigorous/sharp in the factor B and some of the cases need a bit more care. More specifically: - l 98-100: in (2.6) it should be \mathbb{P}(f), not \mathbb{P}_n(l(h,z)), right? - l 144: What happens if BL^*<<r^*<<B^2/n? - l 189: In Theorem 2 the authors have r^* in the upper bound without B. When B is n dependent or very large this could be a problem. The lack of the term 1/B should be also noted when the results are compared to corresponding results in the literature. - 209: where does the B come from in the first bound in (6.1)? - l 269: I guess the authors meant \log V* (or even better |\log V^*|), else it would be negative. Furthermore, (especially in the noisy setting) if V^*\gtrsim n^{\delta}, for some \delta>0 the log n gap is still present...

Correctness: The proofs seem to be correct and there is no methodological contribution.

Clarity: The paper is easy to read, despite being a theoretical work. The authors introduce all of the key concepts and make the manuscript (relatively) self-contained (given the format they do a good job making the paper accessible). However, there are a lot of grammar mistakes/typos, so the whole manuscript has to be very carefully checked. A non-complete list of typos/grammar mistakes are: -l 24: achieves -> achieve - l 50: prelimaries -> prelimaries - l 89: $2^{k+1}r_0$ - l 145: The -> the - l 168: make -> makes - l 169: penalizes-> penalize - l 180: use larger brackets in (5.2) - l 187: are numerical constants - l 193: want -> wants - l 229: strength is not in a correct form - l 232: give -> gives - l 242: both of the - l 249: 1+1\rho->1+\rho - l 263: propose -> proposed - l 264: involking -> invoking - l 267: give -> gives - l 270: inconsistent notation for the variance

Relation to Prior Work: Yes, in my opinion the literature review and connection to other relevant articles are well discussed.

Reproducibility: Yes

Additional Feedback: ******** Update after authors reply ************* The authors have successfully addressed my concerns and I also appreciated that they provided estimators with theoretical guarantees for the parameters V* and L*, making their method more practical. Incorporating the changes suggested by the referees have substantially improved the quality of the manuscript, hence I have decided to raise my assessment grade to 7.

[Author Response · NeurIPS 2020]

We sincerely thank all reviewers for your interest in the paper and the insightful reviews!

**[Responses to R#1] Q1. Can we estimate the bounds from data?** Yes! This is an important question, and we will add

a new corollary stating that all our bounds can be computed from data. In fact, obtaining a preliminary estimate on $\mathcal{L}^*$ is a

step within our two-stage procedure for the variance-dependent rate (see lines 167-172). A very similar analysis answers

this question: we can simply use $\widehat{\mathcal{L}^*} := \mathbb{P}_n \ell(\hat{h}_{\text{ERM}}; z)$ to estimate $\mathcal{L}^*$, and use $\widehat{\mathcal{V}^*} := \mathbb{P}_n[\ell(\hat{h}_{\text{ERM}}; z)^2] - \mathbb{P}_n \ell(\hat{h}_{\text{ERM}}; z)$

to estimate $\mathcal{V}^*$. Furthermore, we can reuse the samples for pure evaluation purpose. Similar to the inequality above line

181, we can bound both $(\widehat{\mathcal{L}^*} - \mathcal{L}^*)^2$ and $|\widehat{V} - \mathcal{V}^*|$ by $O(r^*)$. This precision is enough to rewrite our original bounds by

$\widehat{\mathcal{L}^*}$ and $\widehat{\mathcal{V}^*}$, with other quantities unchanged in order. **Q2. Do we require more efforts to find $\psi$?** We first note that

all previous analyses also require knowing $\psi$ because they rely on knowledge of $r^*$—the fixed point of $B\psi$ (see lines

162-166). When one know the covering numbers, one standard choice of $\psi$ is Dudley's integral used in Examples 1,2.

These illustrate that there is little additional efforts in this identification. **Q3. $\psi$ is dependent on $n$?** Yes, by definition

one must take different $\psi$ for different $n$ (e.g., when taking $n \to \infty$, $\psi$ should be 0). The standard notion "Rademacher

complexity" also depends on fixed $n$. **Notation.** We will clarify that $a \vee b = \max\{a, b\}$ in the preliminaries.

**[Response to R#2] Q2. Can the bounds be described for neural networks?** Yes, though the description contains

eigenvalues that are hard to compute analytically. Our theory *systematically* provides improved problem-dependent rates

as long as one can find good $\psi$ and the class is rich. A line of recent works on infinitely wide neural networks consider

the equivalence between the prediction function found by gradient descent, and the RKHS induced by the "Neural

Tangent Kernel." Many of these works explicitly express the resulting kernel matrix, so our theorems are applicable as

illustrated in Example 3. However, our bounds contain eigenvalues of the kernel matrix, and it is difficult to assess their

decay pattern without further analysis. **Q2. Explain why traditional analysis is optimal for parametric classes?** We

will add the following explanation under line 100. Due to the conceptual proof (2.6), the gap between our result and

the traditional analysis originates from the "sub-root" inequality $\psi(r; \delta)/\sqrt{r} \leq \psi(r^*; \delta)/\sqrt{r^*}$, which is true for all

sub-root $\psi$. This inequality becomes an equality when $\psi(r; \delta) = O(\sqrt{dr/n})$ in the parametric case. However, when

$\mathcal{F}$ is rich, $\psi(r; \delta)/\sqrt{r}$ will be strictly decreasing so that the "sub-root" inequality can be loose (e.g., in Example 1,

$\psi = O(\sqrt{r^{1-\rho}/n})$ so that $\sqrt{\psi(r; \delta)}/\sqrt{r} = O(\sqrt{1/(n \cdot r^\rho)})$). The richer $\mathcal{F}$ is, the more improvement from our theory.

**Writing.** We will reorganize Sections 2-3, and do our best to make the comparison in Section 6 clearer.

**[Responses to R#3]** We are glad to see your appreciation of our machinery! We hope our techniques can become

standard tools to prove adaptive generalization error bounds. **Q1. Trade-off between optimality and practicality?**

There is indeed a trade-off between statistical performance and computation. Similar to majority of previous works

[12, 18, 5], our moment-penalized estimator does not preserve convexity of the population risk, while ERM and the

estimator in [16] do preserve that convexity. In our answer to R#1's Q1, we explain how to compute the bounds from

data. When choosing among different estimators, one can estimate different bounds to decide whether the added price of

optimization results in suitable gains to make it worthwhile. **Q2. Optimality of our results?** A short answer is that, both

our variance-dependent rate and loss-dependent rate exhibit optimal *direct dependence on $n$* when the excess loss class

satisfies standard metric entropy growth conditions. For example, when $\log \mathcal{N}(\varepsilon, \ell \circ \mathcal{H} - \ell \circ h^*; L_2(\mathbb{P}_n)) \leq O(\varepsilon^{-2\rho})$

for a fixed $\rho \in (0, 1)$, both rates match the optimal direct dependence on $n$ given by Dudley's integral. Judging

whether the variance-dependent rate is optimal in all regimes requires constructing a particular class of problems where

$\text{Var}[\ell(h^*; z)] = \mathcal{V}^*$. Although we strongly believe it can be done under a suitable minimax framework, we do not have

a rigorous proof yet. Our loss-dependent rate is proposed for the particular algorithm ERM so the minimax framework

requires further restrictions.

**[Responses to R#4]** Thank you for your throughout reading! The typo list is very helpful, and we will carefully check

the whole manuscript. There are indeed typos on the $B$ factor, but all our rates are actually sharp on $B$. The *generic

correction* is: our loss/variance-dependent rates are $\psi(\mathcal{V}^*; \delta) \vee \frac{r^*}{B} \vee \frac{B \log(1/\delta)}{n}$ and $\psi(B\mathcal{L}^*; \delta) \vee \frac{r^*}{B} \vee \frac{B \log(1/\delta)}{n}$; the

previously best known loss/variance-dependent rates are $\sqrt{\mathcal{L}^* r^*/B} \vee \frac{r^*}{B} \vee \frac{B \log(1/\delta)}{n}$ and $\sqrt{\mathcal{V}^* r^*/B^2} \vee \frac{r^*}{B} \vee \frac{B \log(1/\delta)}{n}$.

**Q1. $\mathbb{P}f$ in inequality (2.6)?** We agree that it is better and clearer to firstly write $\mathbb{P}f$ in the last term of (2.6), then

explains that $\mathbb{P}f$ is close to $\mathbb{P}_n f$ when evaluated at a fixed $f$, and finally contrast this term to the result of the traditional

analysis. **Q2. Comparison in line 144?** In line 207 we explain that for most classes of interests, $r^*$ will be at least of

order $\frac{B^2}{n}$ (this is the order of $r^*$ for a one-dimensional class). We will explain this before line 144 so that we only need

to compare the orders of $B\mathcal{L}^*$ and $r^*$. **On Theorem 2 and line 209.** In the result of Theorem 2 we should correct $r^*$ to

$r^*/B$, and line 209 is correct (see our *generic correction*). Indeed, as $r^*$ is the fixed point of $B\psi$, whenever one want to

take it outside $\psi$, the order should be $\frac{r^*}{B}$. **Correction to VC classes.** That term should be corrected to $\log(B^2/\mathcal{V}^*)$,

and the regime in which we improve all known results is actually $B^2/(\log n)^\alpha \leq \mathcal{V}^*$ with arbitrary fixed $\alpha > 0$. Still,

this is the first result that closes the notorious $O(\log n)$ gap without invoking any further assumptions on $\mathcal{H}$ (e.g., the

complicated "capacity function" assumption in [5]). However, as richer classes exhibit much more improvements, we

will shorten the discussion on VC classes and expand the discussion on kernel classes.

[Meta-Review · NeurIPS 2020]

The reviewers agree that this is an exciting and interesting paper which improves the best-known variance-dependent rates for statistical learning with nonparametric classes, and are all in favor of accepting. I hope the authors will pay attention to the typos and clarifications pointed about by the reviewers and address these in the final version of the paper. As reviewer 4 and the authors' response mention, the point about removing the \log(n) factor about VC classes is subtle, and this paper does not really remove this term unless we make specific assumptions on the value of V*. I would recommend the authors either expand the discussion about this and include a more detailed comparison with prior work, or minimize this claim.